# TABFLEX: Scaling Tabular Learning to Millions with Linear Attention

Yuchen Zeng [* 1 2]   Tuan Dinh [* 3]   Wonjun Kang [4 5]   Andreas C. Müeller [6]

## Abstract

Leveraging the in-context learning (ICL) capability of Large Language Models (LLMs) for tabular classification has gained significant attention for its training-free adaptability across diverse datasets. Recent advances, such as TABPFN, excel in tabular small-scale datasets but struggle to scale for large and complex datasets. Our work enhances the efficiency and scalability of TABPFN for larger datasets by incorporating linear attention mechanisms as a scalable alternative to complexity-quadratic self-attention. Our model, TABFLEX, efficiently handles tabular datasets with thousands of features and hundreds of classes, seamlessly scaling to millions of samples. For instance, TABFLEX processes the `poker-hand` dataset with more than a million samples in just 5 seconds. Our extensive evaluations demonstrate that TABFLEX can achieve over a $2\times$ speedup compared to TABPFN and a $1.5\times$ speedup over XGBoost, outperforming 25 tested baselines in terms of efficiency across a diverse range of datasets. Furthermore, TABFLEX remains highly effective in large-scale datasets, delivering strong performance with significantly reduced computational costs, especially when combined with data-efficient techniques such as dimensionality reduction and data sampling.

## 1. Introduction

Enhancing the applicability of the Transformer architecture (Vaswani et al., 2017) for diverse data modalities beyond textual data and non-language tasks (Achiam et al., 2023; Brown et al., 2020; Bai et al., 2023a; Dubey et al., 2024) has achieved remarkable success (Gemini Team et al., 2023), from vision (Bai et al., 2023b), audio (Chu et al., 2023; 2024) to bio-signals (Wan et al., 2023) and protein sequences (Rives et al., 2019; Hayes et al., 2024). Tabular data, as one of the most fundamental and critical data types in real-world applications – including recommendation systems (Zhang et al., 2019), finance (Arun et al., 2016), and medicine (Johnson et al., 2016) has attracted a great deal of attention and attempts to explore the potential of Transformer-based models, particularly for tabular classification (Arik & Pfister, 2021; Hollmann et al., 2023; Huang et al., 2020; Dinh et al., 2022; Gorishniy et al., 2021). For example, the FT transformer (Gorishniy et al., 2021) converts each sample into a sequence of embeddings to use the transformer to make predictions. TabTransformer (Huang et al., 2020) learns embeddings for categorical features, concatenating them with continuous features. On the other hand, LIFT (Dinh et al., 2022) converts tabular data combined with feature names and task descriptions into textual sentences as input to LLMs. In particular, compared to traditional methods for tabular data such as gradient-boosted trees (Friedman, 2001), these transformer-based methods often suffer from high latency overhead for training and inference, primarily due to their larger model sizes.

The recent TABPFN (Hollmann et al., 2023) addresses the latency limitations of Transformer-based methods by utilizing the in-context learning (ICL) capability (Brown et al., 2020) of LLMs for directly learning a new task from examples without parameter updates, attaining superior efficiency and performance on small-scale datasets. In particular, TABPFN incorporates all training and testing samples into a single prompt and classifies the testing samples in one forward pass, making it highly efficient and effective on simple and small tabular datasets. However, TABPFN faces challenges with complex datasets that require large sample sizes for effective learning, primarily due to the scalability limitations imposed by the quadratic complexity of the attention mechanism, introducing difficulties for both the scalable pretraining and inference processes.

In this work, we address the limitations of the scalability of TABPFN and improve the effectiveness of Transformer-based methods for tabular classification. We first systematically analyze scalable alternatives to attention mechanisms, focusing on state-space models (SSMs) within the Mamba model (Gu & Dao, 2024) and linear attention (Katharopou-

---

[*]Equal contribution [1]Work done during an internship at the Gray Systems Lab, Microsoft [2]University of Wisconsin-Madison [3]University of California San Francisco [4]Furiosa AI [5]Seoul National University [6]Gray System Lab, Microsoft. Correspondence to: Andreas C. Müeller <amueller@microsoft.com>.

*Proceedings of the $42^{nd}$ International Conference on Machine Learning*, Vancouver, Canada. PMLR 267, 2025. Copyright 2025 by the author(s).

los et al., 2020). We find that **(Finding 1)** the inherent causality of SSMs impedes ICL performance compared to non-causal mechanisms. In contrast, **(Finding 2)** linear attention does not suffer from this limitation, maintaining comparable performance with improved computational efficiency. Thus, we develop TABFLEX leveraging linear attention as the attention mechanism, comprising three sub-models where each is optimized for different scenarios and selected based on dataset characteristics (e.g., sample size). TABFLEX efficiently handles tabular datasets with thousands of features and hundreds of classes, scaling to millions of samples. Via the comprehensive evaluation on a diverse range of datasets, we find that **(Finding 3)** TABFLEX consistently achieves competitively high performance with impressive computational efficiency compared to 25 baselines, including TABPFN and XGBoost. Notably, TABFLEX perform highly on `poker-hand` dataset with 1M+ samples in *less than 5 seconds* and attains high accuracies on image datasets such as MNIST (LeCun et al., 2010), Fashion-MNIST (Xiao et al., 2017), and CIFAR-10 (Krizhevsky et al., 2009) in less than one second. Furthermore, our ablation studies suggest that TABFLEX can seamlessly incorporate data-efficient techniques such as dimensionality reduction and data sampling for more computation reduction.

## 2. Related Works

**Transformer-based approaches for tabular classification.** The pioneering TabNet (Arik & Pfister, 2021) applies unsupervised pre-training on masked tabular datasets to infer missing features, enhancing the model's understanding of data and features before supervised learning on feature selection for the final decision boundary. TabTransformer (Huang et al., 2020) proposes handling categorical features by concatenating their contextual embeddings into numerical features. FT-Transformer (Gorishniy et al., 2021) converts samples to embedding sequences using a feature tokenizer for the transformer. LIFT (Dinh et al., 2022) converts each sample into a sentence using a predefined template incorporating the task description and feature names, as the natural input to apply ICL in LLM. TabR (Gorishniy et al., 2024) proposes a retrieval-augmented model with a custom kNN-like component to retrieve and extract signals from the nearest neighbors. BiSHop (Xu et al., 2024) establishes interconnected directional learning modules to process data column-wise and row-wise for tabular learning. XTab (Zhu et al., 2023) utilizes independent featurizers and federated learning to resolve inconsistent column types and quantities.

The widely adopted transformer-based approaches for tabular classification—TABPFN (Hollmann et al., 2023) is trained offline on synthetic datasets derived from previous distributions to perform ICL, allowing efficient inference in small-scale tabular classification tasks. However, it is limited to small tabular classification datasets. To handle it, many concurrent variants are proposed. MixturePFN (Xu et al., 2025) improves scalability by routing new test samples to a pool of scalable prompters using Sparse Mixture of In-Context Prompters, while LoCalPFN (Thomas et al., 2024) proposes retrieving a local subset of task-specific data for efficiently fine-tuning on. Ma et al. (2024) introduce in-context data distillation to optimize TabPFN's context and remove the data size constraint. TuneTable (Feuer et al., 2024) scales TABPFN to large datasets by performing a prefix tuning per dataset. TabPFNv2 (Hollmann et al., 2025) enhances TabPFN's accuracy in low-data regimes (fewer than 10,000 samples), complementing our focus on speed and scalability. Our method is also based on TABPFN, extending its scalability to large datasets while maintaining and improving efficiency by simply replacing the softmax attention with linear attention.

**Attention mechanisms and scalable alternatives.** As Transformers (Vaswani et al., 2017) face the scaling challenge for long sequences due to the quadratic computational and memory complexity, scalable alternatives have been proposed (Orvieto et al., 2023; Sun et al., 2023). While RNNs provide efficient linear-time inference, they struggle with training efficiency and lack the parallelization capabilities of Transformer architectures. Linear attention (Katharopoulos et al., 2020) addresses both concerns by reformulating self-attention as a linear dot-product of kernel feature maps, reducing the computational complexity from quadratic to linear time. Furthermore, causal linear attention can be interpreted as a form of RNN, as it predicts based on a current token and a "hidden state," summarizing information from the previous tokens. State-space models (SSMs) address RNNs' drawbacks by considering linear RNNs with novel algorithms for efficient training (Gu et al., 2021; 2022; Gu & Dao, 2024; Dao & Gu, 2024; Peng et al., 2023; Orvieto et al., 2023; Sun et al., 2023). Dao et al. (2022) identified another bottleneck in attention mechanisms' speed stemming from the relatively slow access to high-bandwidth memory (HBM) in GPUs and proposed FlashAttention (Dao, 2024; Shah et al., 2024) to restructure attention computation to optimize the utilization of high-speed on-chip SRAM while minimizing access to slower HBM, enhancing the efficiency of GPU-based attention operations.

See Section A for an extended discussion of related works.

## 3. Preliminaries

We elucidate key concepts of TABPFN and two prominent scalable attention mechanisms (SSMs and linear attention).

**Implementation of ICL in TabPFN (Hollmann et al., 2023).** Fig. 1 illustrates the design of TABPFN, where each sample is treated as a token, starting with training sam-

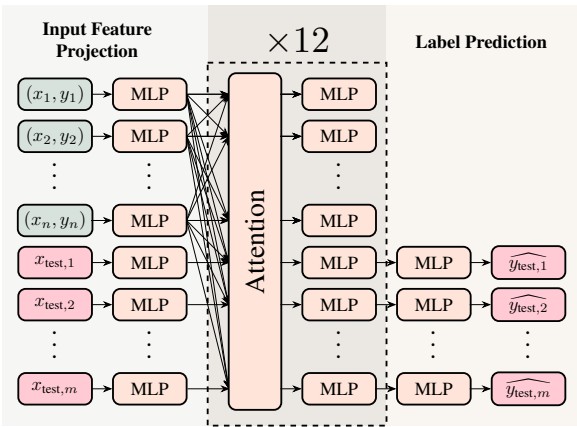

*Figure 1.* **Illustration of TABPFN's for classifying the entire dataset in one forward pass.** In each layer, attention outputs for training sample positions attend to all other training samples, ensuring that predictions are invariant to the order of training samples. Conversely, attention outputs for test sample positions attend only to training samples, ensuring independent predictions for each test instance, unaffected by other test samples. The final classification for each test sample is derived by applying an MLP to the corresponding Transformer output at its respective position.

ples and followed by testing samples. These samples are embedded (features $x$ and labels $y$ for training and only features $x$ for testing samples) with MLPs before being concatenated. Outputs corresponding to training sample positions are computed by attending to all other training samples, while the outputs for test sample positions attend to the training samples — enabling each test prediction to leverage the full training set without being influenced by other test samples. Test predictions are generated by projecting the Transformer outputs at test positions into probability distributions. This implementation is functionally equivalent to standard ICL but significantly more efficient. Standard ICL requires $m$ (number of test samples) separate prompts, each containing all training samples and one test sample, necessitating $m$ prediction passes. A notable feature of TABPFN is the encoder with non-causal attention, allowing outputs within training sample positions to interact freely, rendering the order of training samples inconsequential.

**State-Space Models (SSMs).** The SSM framework is based on a continuous system that transforms a one-dimensional signal $x(t) \in \mathbb{R}$ into $y(t) \in \mathbb{R}$ through an intermediate $H$-dimensional latent state $\boldsymbol{h}(t) \in \mathbb{R}^H$, as shown in (1). Here, $\boldsymbol{B} \in \mathbb{R}^{H \times 1}$ is the input transition vector and $\boldsymbol{A} \in \mathbb{R}^{H \times H}$ is the state transition matrix. The latent state $\boldsymbol{h}(t)$ is then projected into the output $y(t)$ using the output mapping vector $\boldsymbol{C} \in \mathbb{R}^{1 \times H}$. For deep learning applications, discrete $\overline{\boldsymbol{A}}$ and $\overline{\boldsymbol{B}}$ replace continuous $\boldsymbol{A}$ and $\boldsymbol{B}$ through discretization methods, such as zero-order hold. This yields updated hidden state and output equations as shown in (2). While (2) is structured as linear RNN, it can

be reformulated as Convolutional Neural Network (CNN) as (3), enabling efficient and parallelizable training. SSMs address the quadratic time complexity problem w.r.t sequence length, as the output for each new token depends solely on the hidden states and the current token, in contrast to standard attention mechanisms that attend to all previous tokens. Consequently, SSMs operate as a causal mechanism.

$$\boldsymbol{h}'(t) = \boldsymbol{A}\boldsymbol{h}(t) + \boldsymbol{B}x(t), \quad y(t) = \boldsymbol{C}\boldsymbol{h}(t) \tag{1}$$

$$\boldsymbol{h}_t = \overline{\boldsymbol{A}}\boldsymbol{h}_{t-1} + \overline{\boldsymbol{B}}x_t, \quad y_t = \boldsymbol{C}\boldsymbol{h}_t \tag{2}$$

$$\overline{\boldsymbol{K}} = (\boldsymbol{C}\overline{\boldsymbol{B}}, \boldsymbol{C}\overline{\boldsymbol{A}}\,\overline{\boldsymbol{B}}, \dots, \boldsymbol{C}\overline{\boldsymbol{A}}^{t-1}\overline{\boldsymbol{B}}),$$
$$(y_1, \dots, y_t) = (x_1, \dots, x_t) * \overline{\boldsymbol{K}} \tag{3}$$

**Linear attention.** Assume a sequence with length $n \in \mathbb{N}^+$ and embedding size $d \in \mathbb{N}^+$. We first focus on non-causal cases. For the $i$-th position, let $\boldsymbol{q}_i \in \mathbb{R}^d$, $\boldsymbol{k}_i \in \mathbb{R}^d$, and $\boldsymbol{v}_i \in \mathbb{R}^d$ denote the query, key, and value vectors, respectively, where $i = 1, \dots, n$. In softmax attention, the similarity between $\boldsymbol{q}_i$ and $\boldsymbol{k}_j$ for any $i \neq j$ is computed as $\exp\left(\boldsymbol{q}_i^\top \boldsymbol{k}_j\right)$. The attention output at the $i$-th position, denoted as $\boldsymbol{a}_i \in \mathbb{R}^d$, is obtained by averaging the values across all tokens weighted by their similarities. This process requires $O(n)$ complexity, as it necessitates computing similarities with all $n$ tokens. Linear attention reduces this complexity by replacing the similarity computation from $\exp(\boldsymbol{q}_i^\top \boldsymbol{k}_j)$ with $\phi(\boldsymbol{q}_i)^\top \phi(\boldsymbol{k}_j)$, where $\phi : \mathbb{R}^d \to \mathbb{R}^d$ is a feature conversion function. For linear attention outputs (4) across all positions, we identify two common terms: $\sum_{j=1}^n \phi\left(\boldsymbol{k}_j\right) \cdot \boldsymbol{v}_j$ and $\sum_{j=1}^n \phi\left(\boldsymbol{k}_j\right)$, which can be computed once. Consequently, for the linear output at position $i$, we only need to compute $\phi(\boldsymbol{q}_i)$ and multiply it with these two statistics, resulting in $O(1)$ complexity, thus significantly reducing computational demands.

$$\text{(Softmax) } \boldsymbol{a}_i = \frac{\sum_{j=1}^n \exp\left(\boldsymbol{q}_i^\top \boldsymbol{k}_j\right) \cdot \boldsymbol{v}_j}{\sum_{j=1}^n \exp\left(\boldsymbol{q}_i^\top \boldsymbol{k}_j\right)} \tag{4}$$

$$\text{(Linear) } \boldsymbol{a}_i = \frac{\phi\left(\boldsymbol{q}_i\right)^\top \sum_{j=1}^n \phi\left(\boldsymbol{k}_j\right) \cdot \boldsymbol{v}_j}{\phi\left(\boldsymbol{q}_i\right)^\top \sum_{j=1}^n \phi\left(\boldsymbol{k}_j\right)}$$

For causal cases, for position $i$, we replace $\sum_{j=1}^n$ with $\sum_{j=1}^i$, as each token attends only to previous tokens. The statistics then become $\sum_{j=1}^{i-1} \phi\left(\boldsymbol{k}_j\right) \cdot \boldsymbol{v}_j$ and $\sum_{j=1}^{i-1} \phi\left(\boldsymbol{k}_j\right)$, which can be viewed as hidden states in RNNs. Thus, causal linear attention can be conceptualized as a linear RNN, which is also a variant of SSM.

## 4. Architectural Exploration for Scalable Tabular Learning

We analyze State-Space Models and linear attention as attention architecture alternatives to enhance the scalability of TABPFN, focusing on tabular classification with ICL.

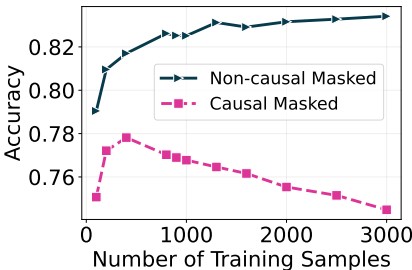

(a) **Effect of causal masking**. When more samples are provided, the non-causal model shows better sample utilization and accuracy, while the causal model's performance plateaus early and declines.

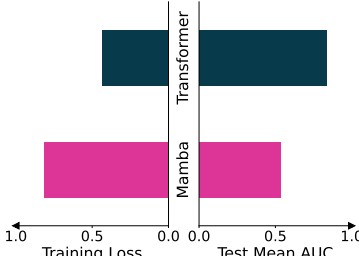

(b) **ICL performance comparison between Mamba and Transformer models**. Results show that Transformer-based models achieve lower training loss and higher AUC across 150 test datasets.

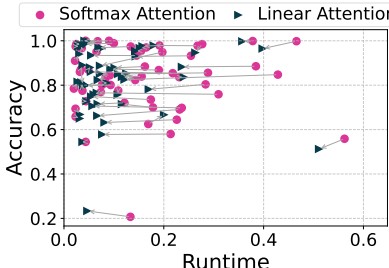

(c) **Accuracy and runtime comparison of softmax and linear attention**. Replacing softmax with linear attention preserves comparable accuracy while significantly reducing runtime.

*Figure 2.* **Impact of model architecture on tabular classification performance.** Please refer to Fig. 8 for a detailed breakdown.

### 4.1. Causal Model vs. Non-Causal Model

Ideally, the order of training samples (i.e., in-context demonstrations) in the prompt should not affect the final prediction. However, SSMs are inherently causal, computing outputs based on new inputs and hidden states derived from previous inputs. This suggests a potential drawback in this context. To validate our hypothesis regarding the suboptimal performance of causal models in ICL, we conduct two experiments: (i) comparing the performance of TABPFN with a modified version of the same model that uses causal attention, and (ii) evaluating TABPFN against both its original version and a model incorporating Mamba (specifically Mamba-II), a leading SSM-based architecture.

**Causal Attention vs. Non-Causal Attention.** Our first experiment compares the ICL capabilities of non-causal and causal attention mechanisms using the same experimental setup as TABPFN, shown in Fig. 2a. We replicate TABPFN's methodology for generating synthetic datasets from priors, training a modified version employing causal attention instead. For the inference, we generate 20 synthetic datasets that maintain a consistent 1000 test samples with varying numbers of training samples. We average the classification accuracy across 20 simulations.

We observe that non-causal attention generally outperforms causal attention. As more training samples are given, the accuracy of the non-causal model continues to improve. In contrast, the causal attention model shows accuracy improvements only within a very small range of training samples, after which performance begins to decline with additional samples. These findings indicate that TABPFN with non-causal attention functions as an effective ICL model, adeptly leveraging context from a large number of samples. Conversely, the same model equipped with causal attention fails to capitalize on the additional data, highlighting the superiority of the non-causal approach in this tabular learning

scenario. Our observation is supported by empirical studies (Ding et al., 2024; Gong et al., 2023), which show that causal attention is suboptimal for ICL. Moreover, most theoretical analyses of ICL assume non-causal attention (Ahn et al., 2023; Bai et al., 2023c).

**Mamba vs. Transformer.** We further investigate whether Mamba, the most popular SSM-based model, is suitable for ICL. We replicate TABPFN's training methodology, substituting the transformer layer with an Mamba layer. To evaluate performance, we test the modified model on the same 150 validation datasets used in the TABPFN study (See Section F.3 for details). Fig. 2b visualizes the training loss and test mean AUC for both methods. The model with Mamba exhibits significantly higher training loss than the original TABPFN, with substantially lower test mean AUC. This experiment with a popular SSM model further demonstrates that SSMs underperform non-causal models in our specified tasks.

### 4.2. Softmax Attention vs. Linear Attention

To address the quadratic complexity of standard attention mechanisms, linear attention has emerged as a popular alternative (Katharopoulos et al., 2020). To investigate its impact on ICL in tabular classification, we replaced TABPFN's attention mechanism with linear attention and trained a model following the same strategy as TABPFN. We then evaluated both TABPFN and this linear attention model on 57 real datasets (used in Table 2 of McElfresh et al. (2023), where TABPFN achieved top performance among 19 methods for tabular classification). Fig. 2c visualizes the test accuracy and runtime. Our results demonstrate that linear attention does not decrease performance and significantly improves speed, making it a suitable method for scaling TABPFN to larger datasets. Finally, in Section B.1, we demonstrate that linear attention significantly outperforms sliding window attention (Beltagy et al., 2020) in our setting.

# 5. TABFLEX: Scaling TabPFN for Large Datasets

Based on the empirical findings presented in Section 4, we identify non-causal linear attention as the optimal candidate to replace standard softmax attention in TABPFN. This section proceeds in two parts: first, we conduct a thorough analysis of the linear attention mechanism to ensure its efficient implementation.; subsequently, we leverage this efficient implementation to train our proposed model, TABFLEX. Our approach aims to enhance the scalability and performance of tabular learning while maintaining computational efficiency.

**Computation Analysis.** Dao et al. (2022) demonstrates that significant wall-clock speedup for softmax attention can be achieved by optimizing the number of memory reads/writes between GPU high bandwidth memory (HBM) and GPU on-chip SRAM. Based on this criterion, Yang et al. (2024) proposed FlashLinearAttention for speeding up *causal* linear attention. This raises a natural question: can we further improve the speed of non-causal linear attention (we omit non-causal when it does not cause further confusion) by reducing the number of memory reads/writes? Our results in Theorem 1 analyze the #HBM access and HBM memory usage of FlashLinearAttention and linear attention, concluding that further optimization is not necessary. In Section C.1, we first propose an HBM-efficient linear attention, and then show that the PyTorch implementation only incurs a marginal increase in terms of #HBM access and HBM memory usage, with FLOPS remaining unchanged. We provide more details, including the analysis of different attention mechanisms and actual memory usage and runtime visualization of these mechanisms in Section C.1. The resulting theorem below demonstrates that the straightforward PyTorch implementation of linear attention achieves linear HBM access, matching the performance of FlashLinearAttention after optimization. Consequently, our model adopts the straightforward implementation of linear attention.

**Theorem 1** (High Bandwidth Memory Efficiency of Linear Attention). *Let $Q, K, V \in \mathbb{R}^{N \times D}$ represent the query, key, and value matrices for a single attention head, where $N$ is the sequence length and $D$ is the embedding size. With any element-wise kernel feature mapping (e.g., $\mathrm{elu}(\cdot) + 1$), both causal FlashLinearAttention (Alg. 2) and non-causal linear attention (Listing 1) require $O(ND)$ HBM accesses, $O(ND)$ HBM memory, and $O(ND^2)$ FLOPS to compute the attention output.*

**TABFLEX.** While TABPFN excels on small, simple datasets with fewer than 100 features and 10 classes, it struggles with more complex tasks, e.g., high-dimensional datasets or those with numerous classes. We aim to extend the use cases by training a model that maintains comparable speed to TABPFN while offering reasonable performance

---

**Algorithm 1** Conditional Model Selection

**Input :** A dataset $\mathcal{D}$ with $n$ instances of $d$ features

1  // Large dataset with few features  **if** $n \geq 3K$ *and* $d \leq 100$ **then**
2  $\quad$ | **return** TABFLEX-L100$(\mathcal{D})$
3  // High-dimensional datasets  **else if** $d > 100$ *or* $(d/n \geq 0.2$ *and* $n \geq 3K)$ **then**
4  $\quad$ | **if** $d \leq 1000$ **then**
5  $\quad\quad$ | **return** TABFLEX-H1K$(\mathcal{D})$
6  $\quad$ | **else**
7  $\quad\quad$ Apply random projection to $\mathcal{D}$ to reduce the number of features to 1000, yielding $\mathcal{D}'$ **return** TABFLEX-H1K$(\mathcal{D}')$
8  // Small datasets  **else**
9  $\quad$ | **return** TABFLEX-S100$(\mathcal{D})$

---

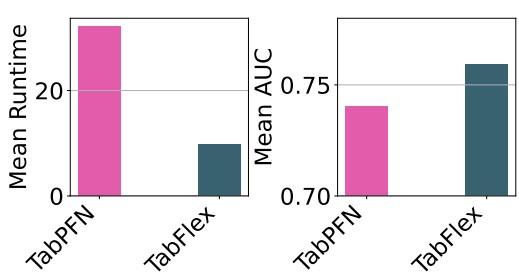

*Figure 3.* Runtime and AUC comparison of TABPFN and TABFLEX on validation datasets.

across a broader spectrum of datasets. Since models trained on large, high-dimensional datasets often struggle in small regions due to optimization issues, we introduce three specialized models to address this limitation.

- **TABFLEX-S100** is trained on prompts with 1152 length (same as TABPFN), 100 features, and 10 classes. This is optimized for low-dimensional datasets. 'S' denotes standard configuration, '100' indicates feature capacity.

- **TABFLEX-L100** utilizes prompts of 50K length, 100 features, and 10 classes. This is designed for large, low-dimensional datasets. 'L' signifies a larger sample size, and '100' represents feature count.

- **TABFLEX-H1K** employs prompts of 50K length, 1K features, and 100 classes. This is suited for large, high-dimensional datasets. 'H' indicates high-dimensional capabilities, and '1K' denotes 1K features.

Additional training details, including training loss, hyperparameters, and other relevant information, are provided in Section C.2. Our code is available at https://github.com/microsoft/ticl.

We apply the conditional model selection strategy, shown in Alg. 1, to select the model based on the target dataset's size

and dimensionality, ensuring optimal performance across diverse data characteristics. The decision thresholds align with the training regimes of the models. TABFLEX-S100, sharing TabPFN's training setup but with an updated architecture, is deployed similarly ($n \leq 3K, d \leq 100$). TABFLEX-L100, trained on low-dimensional ($d \leq 100$) but larger datasets, is used for longer sequences ($n \geq 3K, d \leq 100$). TABFLEX-H1K, trained on high-dimensional data, is assigned to handle those cases accordingly. We note that performance is not highly sensitive to the chosen decision boundaries, supported by our results in Section C.4.

In Fig. 3, we visualize the mean runtime and mean AUC comparison of TABPFN and TABFLEX on the validation datasets, comprising 40 datasets with varying sample sizes (up to 100K), dimensions (up to 3K), and number of classes (up to 100). Detailed information about these datasets is provided in Section C.3. Our analysis reveals that TABFLEX not only exhibits superior performance but also demonstrates faster execution times compared to TABPFN.

## 6. Performance Evaluation of TABFLEX

We evaluate TABFLEX's performance and speed across 115 OpenML tabular datasets (Vanschoren et al., 2013). See Section D.2 for the complete list of baselines, and a detailed description of the models' implementation.

### 6.1. Experimental Setup

Unless otherwise stated, we follow the identical experiment setup of McElfresh et al. (2023) for baseline benchmarking.

**Datasets.** For classification tasks, we consider datasets of two difficulty levels. For simpler tasks, we use two collections of datasets—98 and 57 in total—originally reported in Tables 1 and 2 of McElfresh et al. (2023), which are characterized by smaller sample sizes and lower feature dimensions. For more challenging tasks, we evaluate the methods on the TabZilla hard benchmark, which includes 36 difficult datasets—11 of which are high-dimensional (100–2000 features) and large-scale ($\geq 50K$ instances). Detailed dataset information, including names and characteristics, is provided in Section D.1, with additional datasets and results available in Section D.4.

**Baselines.** We evaluate our approach against a comprehensive set of baselines, as considered by McElfresh et al. (2023), including (i) four classical methods, e.g., Random Forest (Liaw et al., 2002), (ii) three Gradient Boosted Decision Trees (GBDT) methods, e.g., XGBoost (Chen & Guestrin, 2016), (iii) ten Non-Transformer Neural Network (Non-TF NN) methods, e.g., SAINT (Somepalli et al., 2021), (iv) four Transformer (TF) methods, e.g., TABPFN, with two recent methods designed for scaling tabular classifica-

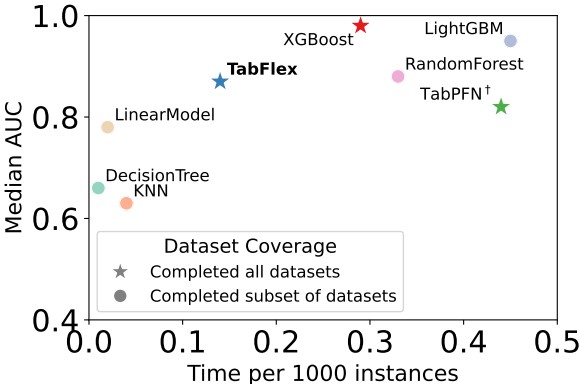

*Figure 4.* **Visualization of tested methods with processing times under 0.5 seconds per 1000 instances on the TabZilla hard benchmark.** We report the median AUC across the completed datasets, as several methods completed only a subset of the datasets. Compared to other methods (XGBoost and TABPFN) that successfully ran on all datasets, TABFLEX achieves a 2× speedup while maintaining relatively good performance.

tion: TuneTables (Feuer et al., 2024), a TF method, and HyperFast (Bonet et al., 2024), a Non-TF NN method.

### 6.2. Evaluation on Simple Datasets

We evaluate models on two sets of data: 98 simple datasets from Table 1 and 57 small datasets from Table 2 of McElfresh et al. (2023). The results are reported in Table 11 (Section D.3) and Table 1, respectively. For each dataset, we consider ten different train/test splits, computing the score mean and standard deviation, as well as the total runtime per 1000 instances. We then calculate the median and mean of these values across the entire set of datasets: 98 simple datasets for Table 11 and 57 small datasets for Table 1. Algorithms are ranked based on AUC and time. Our results demonstrate that TABFLEX achieves nearly identical performance to TABPFN on small, simple datasets while offering more than a 2× speedup. Compared to faster methods, such as Decision Tree and Linear Model in Table 11, and Decision Tree, Linear Model, LightGBM, and KNN in Table 1, their performance is significantly inferior to TABFLEX.

### 6.3. Evaluation on Hard Datasets

In this experiment, we compare TABFLEX to baselines on the TabZilla hard benchmark (McElfresh et al., 2023), which includes 36 datasets. However, due to the challenging nature of the datasets in the TabZilla hard benchmark, many baselines fail to execute successfully. In Fig. 4, we visualize the Median AUC and the runtime per 1000 instances across the 36 datasets, with methods that successfully executed on all datasets marked as stars, and methods that failed to execute on some datasets marked as circles. This figure focuses on efficient methods, excluding those slower

| Algorithm | Class | Mean AUC | | Std. AUC | | Time / 1000 inst. | |
|---|---|---|---|---|---|---|---|
| | | median | mean | mean | median | median | mean |
| TabPFN (Hollmann et al., 2023) | TF | 0.97 | 0.90 | 0.21 | 0.15 | 0.82 | 1.04 |
| **TABFLEX (Ours)** | TF | 0.96 | 0.89 | 0.22 | 0.16 | 0.29 | 0.48 |
| CatBoost (Prokhorenkova et al., 2018) | GBDT | 0.95 | 0.89 | 0.23 | 0.16 | 2.59 | 19.51 |
| ResNet (He et al., 2016) | Non-TF NN | 0.93 | 0.84 | 0.24 | 0.16 | 13.90 | 23.40 |
| SAINT (Somepalli et al., 2021) | TF | 0.93 | 0.84 | 0.24 | 0.20 | 173.63 | 195.16 |
| RandomForest (Liaw et al., 2002) | Classical | 0.92 | 0.86 | 0.24 | 0.17 | 0.45 | 0.61 |
| XGBoost (Chen & Guestrin, 2016) | GBDT | 0.91 | 0.86 | 0.24 | 0.18 | 0.49 | 0.95 |
| HyperFast (Bonet et al., 2024) | Non-TF NN | 0.91 | 0.83 | 0.22 | 0.17 | 64.38 | 136.74 |
| DANet (Chen et al., 2022) | Non-TF NN | 0.89 | 0.80 | 0.25 | 0.19 | 67.70 | 78.21 |
| SVM (Cortes, 1995) | Classical | 0.87 | 0.75 | 0.28 | 0.22 | 0.71 | 87.84 |
| NODE (Popov et al., 2019) | Non-TF NN | 0.86 | 0.80 | 0.24 | 0.18 | 157.18 | 194.07 |
| DeepFM (Guo et al., 2017) | Non-TF NN | 0.86 | 0.79 | 0.28 | 0.27 | 5.48 | 5.95 |
| FTTransformer (Gorishniy et al., 2021) | TF | 0.84 | 0.78 | 0.25 | 0.21 | 25.40 | 33.34 |
| LightGBM (Ke et al., 2017) | GBDT | 0.83 | 0.76 | 0.28 | 0.21 | 0.25 | 0.67 |
| MLP-rtdl (Gorishniy et al., 2021) | Non-TF NN | 0.83 | 0.74 | 0.26 | 0.20 | 12.65 | 22.97 |
| LinearModel (Cox, 1958) | Classical | 0.81 | 0.71 | 0.27 | 0.21 | 0.05 | 0.06 |
| TuneTables (Feuer et al., 2024) | TF | 0.80 | 0.72 | 0.32 | 0.24 | 53.48 | 113.49 |
| STG (Yamada et al., 2020) | Non-TF NN | 0.79 | 0.67 | 0.29 | 0.23 | 18.46 | 21.26 |
| TabTransformer (Huang et al., 2020) | TF | 0.79 | 0.64 | 0.24 | 0.16 | 19.04 | 32.84 |
| MLP (Rumelhart et al., 1986) | Non-TF NN | 0.72 | 0.65 | 0.29 | 0.25 | 17.83 | 27.67 |
| DecisionTree (Quinlan, 1986) | Classical | 0.63 | 0.55 | 0.35 | 0.31 | 0.01 | 0.02 |
| KNN (Cover & Hart, 1967) | Classical | 0.62 | 0.56 | 0.30 | 0.25 | 0.03 | 0.03 |
| TabNet (Arik & Pfister, 2021) | TF | 0.56 | 0.50 | 0.42 | 0.40 | 34.66 | 42.09 |
| VIME (Yoon et al., 2020) | Non-TF NN | 0.49 | 0.48 | 0.37 | 0.27 | 18.43 | 20.11 |
| NAM (Agarwal et al., 2021) | Non-TF NN | 0.33 | 0.38 | 0.38 | 0.31 | 147.30 | 341.58 |

*Table 1.* **Performance of algorithms across 57 datasets of size less than or equal to 1250 (used in Table 2 of McElfresh et al. (2023)).** The reported AUC values are normalized. The "Time/1000 inst." column represents the combined training and test time for all datasets, divided by the total number of samples. Notably, TABFLEX achieves top-2 performance, with significantly faster runtimes compared to baselines of similar performance, and a 2× speedup relative to TABPFN.

than 0.5 seconds per 1000 instances. We observe that only TABFLEX, TABPFN, and XGBoost successfully run on all datasets. Notably, TABFLEX is faster and achieves better performance than TABPFN, and is faster than XGBoost while sacrificing only a small margin of performance.

Next, we focus on 11 high-dimensional and large datasets within the TabZilla hard benchmark. Since most baselines do not obtain complete results for all datasets, instead of comparing TABFLEX to a specific baseline, we report the 5th-best AUC and 5th-best runtime, using these values to summarize the general performance distribution of the baselines. The results are presented in Table 2. We observe that, for these datasets, TABFLEX substantially outperforms TABPFN. While TABPFN follows McElfresh et al. (2023)'s strategy of using only 3000 training samples, TABFLEX utilizes all available training data, achieving superior performance with comparable or slightly higher processing times. TABFLEX exhibits competitive performance among baselines while maintaining high efficiency. Notably, on large datasets with more than 50K instances, TABFLEX is significantly faster than the baselines. For instance, on the largest

dataset, *poker-hand*, containing over one million samples, TABFLEX significantly outperforms other baselines, classifying all samples in just 4.88 seconds, while the fifth fastest method requires more than 500 seconds.

### 6.4. Extension to Regression Tasks

So far, we have presented our evaluation on classification tasks. For the extension to regression, a simple workaround is to convert the task into classification by discretizing the target range into bins. Shown in Table 3, we apply this technique for regression data with numerical features from Grinsztajn et al. (2022), where targets are discretized into 10 and 100 uniform bins. For baselines, we use linear regression and the XGBoost Regressor (100 estimators, max depth 6), both with default parameters from the `scikit-learn` package. Although regression is not the primary goal of TABFLEX, it demonstrates reasonable performance.

Furthermore, we compare TABFLEX with TuneTables (Feuer et al., 2024) using their setup in Sec. D.5 and extend TABFLEX to image datasets, shown in Section D.6.

| Dataset | #Classes | #Features | #Instances | AUC | | | Time (seconds) | | |
|---|---|---|---|---|---|---|---|---|---|
| | | | | 5th Best | TABPFN | TABFLEX | 5th Best | TABPFN | TABFLEX |
| SpeedDating | 2 | 120 | 8378 | 0.86 | 0.55 | 0.85 | 1.58 | 1.58 | 1.89 |
| higgs | 2 | 28 | 98050 | 0.79 | 0.72 | 0.76 | 3.46 | 2.82 | 4.92 |
| cnae-9 | 9 | 856 | 1080 | 1.00 | 0.48 | 0.96 | 0.51 | 0.51 | 3.80 |
| albert | 2 | 78 | 425240 | 0.71 | 0.69 | 0.70 | 33.98 | 9.39 | 13.46 |
| audiology | 24 | 69 | 226 | 0.92 | 0.82 | 0.81 | 0.13 | 0.23 | 0.26 |
| jasmine | 2 | 144 | 2984 | 0.86 | 0.70 | 0.86 | 0.68 | 1.27 | 0.99 |
| nomao | 2 | 118 | 34465 | 0.99 | 0.76 | 0.99 | 4.03 | 1.82 | 5.34 |
| Bioresponse | 2 | 1776 | 3751 | 0.85 | 0.50 | 0.75 | 2.49 | 1.29 | 12.38 |
| MiniBooNE | 2 | 50 | 130064 | 0.98 | 0.98 | 0.97 | 10.80 | 3.19 | 7.22 |
| airlines | 2 | 7 | 539383 | 0.70 | 0.63 | 0.64 | 6.53 | 9.73 | 4.20 |
| poker-hand | 10 | 10 | 1025009 | 0.54 | 0.72 | 0.84 | 504.52 | 15.36 | 4.88 |

*Table 2.* **Performance comparison of TABFLEX, TABPFN, and other baselines on large, high-dimensional datasets from the TabZilla hard benchmark (McElfresh et al., 2023).** Baseline results are summarized by the 5th highest AUC and the 5th lowest runtime for each dataset. TABFLEX significantly outperforms TABPFN on these datasets, achieving comparable performance to other baselines while maintaining exceptional speed.

| Dataset | TABFLEX | Linear Regression | XGBoost |
|---|---|---|---|
| cpu_act | 0.9622 | 0.7661 | 0.9872 |
| pol | 0.7770 | 0.4471 | 0.9876 |
| elevators | 0.7386 | 0.8336 | 0.8984 |
| wine_quality | 0.1966 | 0.2842 | 0.4398 |
| Ailerons | 0.7284 | 0.8137 | 0.8272 |
| houses | 0.6803 | 0.6496 | 0.8469 |
| house_16H | 0.2519 | 0.1708 | 0.5276 |
| diamonds | 0.9085 | 0.9213 | 0.9477 |
| Brazilian_houses | 0.8943 | 0.3459 | 0.9828 |
| Bike_Sharing_Demand | 0.3796 | 0.3291 | 0.6995 |
| nyc-taxi-green-dec-2016 | 0.1547 | 0.3109 | 0.5732 |
| house_sales | 0.6656 | 0.7375 | 0.8732 |
| sulfur | 0.4026 | 0.3068 | 0.7497 |
| medical_charges | 0.8173 | 0.8118 | 0.9790 |
| MiamiHousing2016 | 0.8112 | 0.7302 | 0.9306 |
| superconduct | 0.6867 | 0.7169 | 0.9086 |
| yprop_4_1 | 0.0000 | 0.0449 | 0.0000 |
| abalone | 0.3689 | 0.4622 | 0.5125 |

*Table 3.* **Comparison of performance across regression tasks for TABFLEX, Linear Regression, and XGBoost.** Regression datasets are from Grinsztajn et al. (2022). To extend TABFLEX to regression, we discretize the target variable into 10 and 100 uniform bins and use the better-performing setting, converting the task to classification. Despite being designed for classification, TABFLEX delivers reasonable performance on regression tasks.

# 7. Ablation Studies

We conduct ablation studies, including a fine-grained comparison with XGBoost and the integration of other data-efficient techniques. Performance and runtime trends with respect to training sample sizes, along with detailed experimental setups, are provided in Section E.

## 7.1. Fine-Grained Comparison with XGBoost

In Fig. 4, we observe a larger performance gap between TABFLEX and XGBoost compared to the simpler datasets

shown in Table 1. To better understand this discrepancy, we conduct a more fine-grained comparison between TABFLEX and XGBoost using synthetic datasets. XGBoost is configured with a tree depth of 3 and 20 estimators to balance speed and accuracy. We evaluate the accuracy-runtime tradeoff across varying feature dimensions $(100, 200, 400, 600, 800, 1000)$ and training sample sizes $(1000, 2000, \dots, 12000)$, averaging results over 20 synthetic datasets with diverse distributions. TABFLEX consistently outperforms XGBoost in both accuracy and inference time when the feature dimensionality is below 800. As the number of features increases, the performance gap narrows, and XGBoost eventually surpasses TABFLEX at 800 features. Nevertheless, TABFLEX achieves a stronger overall tradeoff across most settings.

## 7.2. Incorporating Data-Efficient Techniques: Dimensionality Reduction and Data Sampling

Since TABFLEX utilizes the ICL for prediction, reducing the complexity of the data can further improve the inference efficiency. We combine TABFLEX with two data-efficient techniques (feature dimension reduction and training data sampling) and investigate how they affect the balance between efficiency and classification performance. See Section E.1 for detailed setup and datasets.

First, for dimensionality reduction, we apply three techniques: Principal components analysis (PCA) (Maćkiewicz & Ratajczak, 1993), Singular Value Decomposition (SVD), and random linear projection (Vempala, 2005). Fig. 6 presents the performance and latency of TABFLEX where feature dimensions are varied. Datasets are selected from Table 9 where feature dimensions are greater than 100. On average, dimensions can be reduced to 10% to reduce latency more than 2 times while AUC scores remain the same

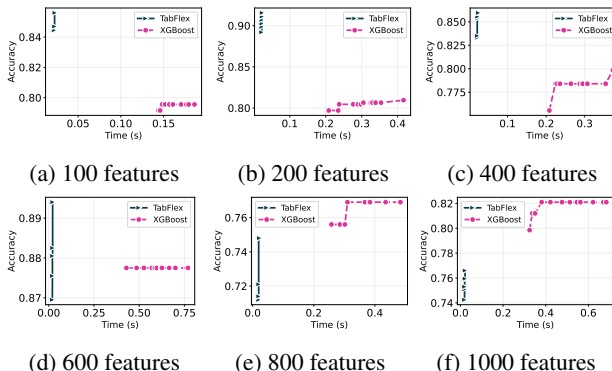

(a) 100 features    (b) 200 features    (c) 400 features

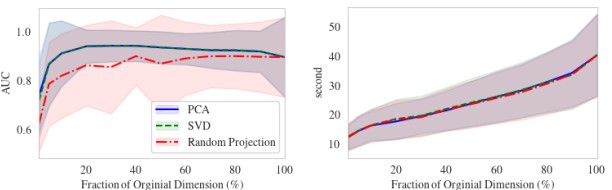

(d) 600 features    (e) 800 features    (f) 1000 features

*Figure 5.* **Accuracy-runtime tradeoff of TABFLEX and XG-Boost across feature dimensions** $(100, 200, 400, 600, 800, 1000)$ **and training sample sizes** $(1000, 2000, \dots, 12000)$**.** Results are averaged over 20 synthetic datasets with diverse data distributions. TABFLEX consistently achieves better performance and faster inference than XGBoost when the number of features is below 800. As dimensionality increases, the gap diminishes, with XGBoost overtaking TABFLEX at 800 features.

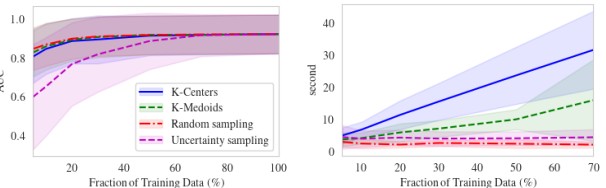

*Figure 6.* **Varying feature dimension with dimensionality reduction methods.** Results are measured on a set of datasets whose number of features is greater than 100. Dimension of features can be reduced up to 90% to preserve the performance (left) with inference being $2\times$ faster (right).

or even better than the original predictions (100%).

Similarly, we conduct ablation studies on training data size with different sampling methods (K-centers, K-medoids, uncertainty sampling, and random sampling), shown in Fig. 7. The tested datasets are selected from Table 9 where the data size is greater than 1000 instances and the feature dimension is lower than 100. Fig. 7 illustrates that the original performance can be preserved with only 20% of training data while the latency can be significantly reduced.

## 8. Conclusion & Discussion

**Conclusion.** To extend TABPFN for ICL on larger and more challenging tabular classification tasks, in this paper, we conduct a comprehensive exploration of scalable alternatives to attention, ultimately selecting non-causal linear attention. Through computational analysis for algorithmic optimization of the implementation of linear attention, we develop our model, TABFLEX. We demonstrate that TABFLEX achieves comparable performance to TABPFN on small

*Figure 7.* **Varying training data with sampling methods.** Results are measured on a set of datasets whose data size is greater than 1000 and whose dimension is lower than 100. For most datasets, only 20% of training data may be required to preserve the performance (left), while significantly reducing the latency (right).

datasets with more than $2\times$ speedup, while outperforming most other baselines with significantly reduced computational time. Moreover, TABFLEX significantly outperforms TABPFN on larger and more complex datasets, becoming much faster than most other baselines on datasets larger than 100K samples, while maintaining performance on par with state-of-the-art methods. We posit that TABFLEX further elevates the performance ceiling of neural network-based models on tabular classification tasks.

**Limitations & Future Works.** While our method achieves fast inference and competitive performance on datasets with around two thousand features, scaling to even larger feature spaces remains a compelling research direction. In this paper, we also explored adapting TABFLEX to image classification tasks on small-scale datasets such as MNIST and CIFAR-10. Extending this approach to large-scale image classification could broaden its applicability, especially given its extremely fast inference and ability to generate predictions for all test samples simultaneously. Further extending TABFLEX to other modalities such as audio and video is another promising direction. This may require new strategies for synthetic data generation for pre-training, along with systematic analyses of the effects of architectural hyperparameters such as depth and embedding size. These efforts would help optimize the model for high-dimensional inputs and enhance its generalization across domains. Currently, our focus on tabular tasks is limited to classification. We attempted a naive extension to regression by discretizing the target range into bins and treating it as a classification problem. A promising future direction is to extend TABFLEX to regression tasks using a more principled approach, such as training on regression-specific synthetic data. Lastly, TabPFNv2 is a concurrent work that further improves the performance of TABPFN. Investigating how incorporating linear attention might impact TabPFNv2 is also an interesting question for future research.

## Impact Statement

This research tackles a key limitation in applying TABPFN to tabular classification tasks: scalability. By introducing TABFLEX, which leverages linear attention, it enables efficient processing of tabular datasets with millions of samples and thousands of features. The primary impact lies in making in-context learning feasible for a broader range of large-scale tabular problems, which is particularly useful in domains such as finance and recommendation systems. Furthermore, the successful integration of linear attention in TABFLEX lays the groundwork for future studies on efficient attention mechanisms and model architectures for tabular and other non-NLP domains, as supported by promising preliminary results in image classification.

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

# Appendix

# A. Extended Section 2: Related Works

**Classical Machine Learning Approaches for Tabular Classification.** Classical machine learning algorithms have long been the foundation of tabular data classification. These methods include k-Nearest Neighbors (KNN) (Cover & Hart, 1967), Logistic Regression (Cox, 1958), Decision Trees (Quinlan, 1986), and Support Vector Machines (SVM) (Cortes, 1995). These classical models, while effective, often struggle to handle complex, high-dimensional tabular datasets, motivating the development of more sophisticated approaches.

**Gradient-Boosting Decision Trees for Tabular Classification** Gradient-boosting decision trees (GBDTs) (Friedman, 2001) have emerged as a cornerstone in tabular classification, owing to their exceptional ability to capture intricate patterns in structured data. GBDTs refine their outputs to minimize errors by iteratively combining predictions from weak learners, resulting in high predictive accuracy. XGBoost (Chen & Guestrin, 2016) introduced weighted quantile sketching, advanced regularization techniques, and sparsity awareness, achieving state-of-the-art performance. LightGBM (Ke et al., 2017), a computationally efficient GBDT implementation, employs Gradient-based One-Side Sampling and a leaf-wise tree growth strategy. CatBoost (Prokhorenkova et al., 2018) leverages symmetric trees and introduces ordered boosting, with a particular emphasis on effectively handling categorical features. These advancements have rendered GBDTs not only powerful but also versatile tools in the domain of tabular data, dominating tabular classification in terms of both speed and performance until the advent of TABPFN.

**Transformer-based Approaches for Tabular Classification.** Recent years have witnessed numerous attempts to employ Transformers for tabular classification (Arik & Pfister, 2021; Huang et al., 2020; Gorishniy et al., 2021; Dinh et al., 2022; Hollmann et al., 2023). These methods utilize Transformers in diverse ways to tackle tabular data. TabNet (Arik & Pfister, 2021), one of the pioneering efforts, applies unsupervised pre-training on masked tabular datasets to infer missing features, thereby enhancing the model's understanding of datasets and features. It then performs supervised learning on feature selection to obtain the final decision boundary, akin to decision trees. Huang et al. (2020) introduced TabTransformer, which leverages Transformers to better handle categorical features by concatenating their contextual embeddings with numerical features. While TabTransformer processes categorical and continuous features separately, SAINT (Somepalli et al., 2021) projects both feature types into a shared embedding space before passing them through transformer blocks, thereby enhancing overall performance. FT-Transformer (Gorishniy et al., 2021) introduces a feature tokenizer to convert each example into a sequence of embeddings, enabling Transformers to process tabular datasets and make predictions. LIFT (Dinh et al., 2022) utilizes a pre-trained language model with parameter-efficient fine-tuning, incorporating task descriptions and converting each sample into a complete sentence with feature names in the prediction prompt. TabR (Gorishniy et al., 2024) proposes a retrieval-augmented model with a custom kNN-like component to retrieve and extract signals from the nearest neighbors. BiSHop (Xu et al., 2024) establishes interconnected directional learning modules to process data column-wise and row-wise for tabular learning. XTab (Zhu et al., 2023) utilizes independent featurizers and federated learning to resolve inconsistent column types and quantities.

TABPFN (Hollmann et al., 2023) is trained offline on synthetic datasets derived from prior distributions and performs ICL rather than additional parameter tuning for a given dataset, enabling it to solve small tabular classification tasks within seconds. Prior to our work, TuneTable (Feuer et al., 2024) extended TABPFN to scale to large datasets by performing prefix-tuning for each dataset to achieve better performance. MixturePFN (Xu et al., 2025) improves scalability by routing new test samples to a pool of scalable prompters using Sparse Mixture of In-Context Prompters, while LoCalPFN (Thomas et al., 2024) proposes retrieving a local subset of task-specific data for efficiently fine-tuning on. Ma et al. (2024) introduce in-context data distillation to optimize TabPFN's context and remove the data size constraint. TuneTable (Feuer et al., 2024) scales TABPFN to large datasets by performing a prefix tuning per dataset. TabPFNv2 (Hollmann et al., 2025) enhances TabPFN's accuracy in low-data regimes (fewer than 10,000 samples), complementing our focus on speed and scalability. Our method is also based on TABPFN, extending its scalability to large datasets while maintaining and improving efficiency by simply replacing the softmax attention with linear attention.

**Attention Mechanisms and Scalable Alternatives.** While attention in Transformers (Vaswani et al., 2017) is central to the strong performance of language models, it encounters scaling challenges for long sequences due to its quadratic computational and memory complexity. To overcome these limitations, several scalable alternatives have been proposed (Gu & Dao, 2024; Dao & Gu, 2024; Katharopoulos et al., 2020; Peng et al., 2023; Orvieto et al., 2023; Sun et al., 2023), all aiming to achieve subquadratic time complexity. In contrast, classical RNNs provide the advantage of efficient linear-time inference but suffer from limitations in training efficiency, lacking the parallelization capabilities of Transformer architectures. Linear

attention (Katharopoulos et al., 2020) addresses both concerns by reformulating self-attention as a linear dot-product of kernel feature maps, reducing the computational complexity from quadratic to linear time. Additionally, causal linear attention can be interpreted as a form of RNN, as the model makes predictions based on a current token and a "hidden state," which summarizes information from the previous tokens. State-space models (SSMs), another popular variant of RNNs, address the drawbacks of classical RNNs by considering linear RNNs and proposing novel algorithms for efficient training (Gu et al., 2021; 2022; Gu & Dao, 2024; Dao & Gu, 2024; Peng et al., 2023; Orvieto et al., 2023; Sun et al., 2023).

Dao et al. (2022) identified that another bottleneck in attention mechanisms' speed stems from the relatively slow access to high-bandwidth memory (HBM) in GPUs. To address this limitation, FlashAttention (Dao et al., 2022; Dao, 2024; Shah et al., 2024) restructures attention computation to optimize the utilization of high-speed on-chip SRAM while minimizing access to slower HBM, thereby enhancing the efficiency of GPU-based attention operations. FlashAttention strategically balances computational efficiency against memory bandwidth efficiency. Although the computational complexity in terms of sequence length remains quadratic, the optimizations introduced by FlashAttention significantly accelerate attention computation in wall-clock time.

**Non-Transformer Neural Network-based Approaches for Tabular Classification.** Non-Transformer neural networks, such as Multi-Layer Perceptrons (MLP) (Rumelhart et al., 1986), were explored for tabular classification long before Transformer-based methods, but their performance was limited. In recent years, several novel neural network techniques have been developed for this task, including ResNet (He et al., 2016), DANet (Chen et al., 2022), NODE (Popov et al., 2019), DeepFM (Guo et al., 2017), STG (Yamada et al., 2020), VIME (Yoon et al., 2020), and NAM (Agarwal et al., 2021). DeepFM (Guo et al., 2017) employs a factorization machine-based neural network to learn from categorical data. Drawing inspiration from CatBoost, Popov et al. (2019) presents a novel neural network architecture designed specifically for tabular data, named Neural Oblivious Decision Ensembles (NODE). While self- and semi-supervised learning have demonstrated effectiveness in the domains of computer vision and natural language processing, Yoon et al. (2020) proposed Value Imputation and Mask Estimation (VIME), which represents the first attempt to address tabular tasks using a self- and semi-supervised learning framework. (Agarwal et al., 2021) proposed the Neural Additive Model (NAM), an interpretable neural network that maintains strong performance on tabular data. Yamada et al. (2020) proposed a feature selection method using stochastic gates (STG), which is a neural network-based and effective approach for tabular data. Chen et al. (2022) designed an abstract layer, a specialized neural component for tabular data, and proposed Deep Abstract Networks (DANets) by stacking these layers.

Some approaches even replace Transformers with SSMs for tabular learning (Ahamed & Cheng, 2024; Thielmann et al., 2024). However, these methods require training on a per-dataset basis, leading to high computational costs, and they are generally slower than GBDTs for tabular classification tasks.

**Linear Attention for In-Context Learning.** Although linear attention has been reported to underperform in some language modeling tasks (You et al., 2024; Zhang et al., 2024; Qin et al., 2022), recent theoretical work demonstrates its effectiveness in in-context learning scenarios, where it can emulate gradient descent to achieve learning during inference (Ahn et al., 2023).

## B. Supplement to Section 4: Architectural Exploration for Scalable Tabular Learning

### B.1. Comparison with Other Attention Variant

In addition to the broad categories of all linear RNN variant models we studied in this paper, we also consider another mechanism that enjoys linear complexity: sliding window attention (Beltagy et al., 2020). We show that TABFLEX achieves significantly better performance.

| Method | #Class | #Features | #Instances | Sliding Window | Linear (Ours) |
|---|---|---|---|---|---|
| Poker-Hand | 10 | 10 | 1,025,009 | 0.48 | **0.84** |
| Airlines | 2 | 7 | 539,383 | 0.48 | **0.64** |
| Higgs | 2 | 28 | 98,050 | 0.39 | **0.76** |

*Table 4.* Performance comparison of TABFLEX with Sliding Window attention.

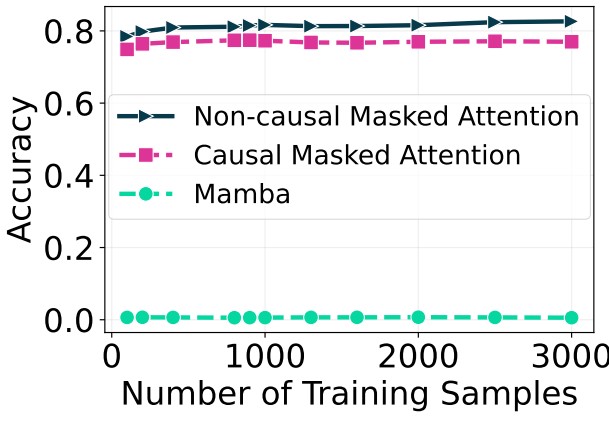

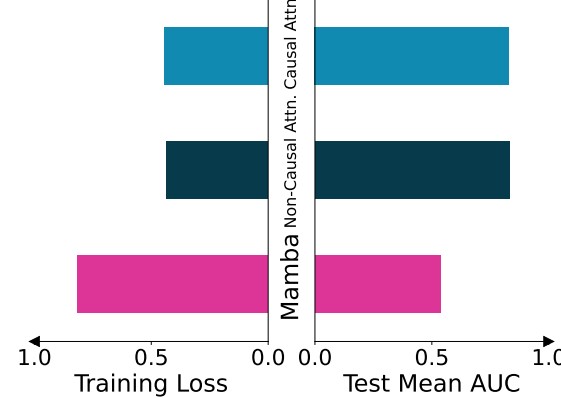

(a) **Effect of causal masking**. When more samples are provided, the non-causal model shows better sample utilization and accuracy, while the causal model's performance plateaus early and declines. The Mamba variant performs poorly.

(b) **ICL performance comparison between Mamba and Transformer models**. Results show that Transformer-based models achieve lower training loss and higher AUC across 150 test datasets.

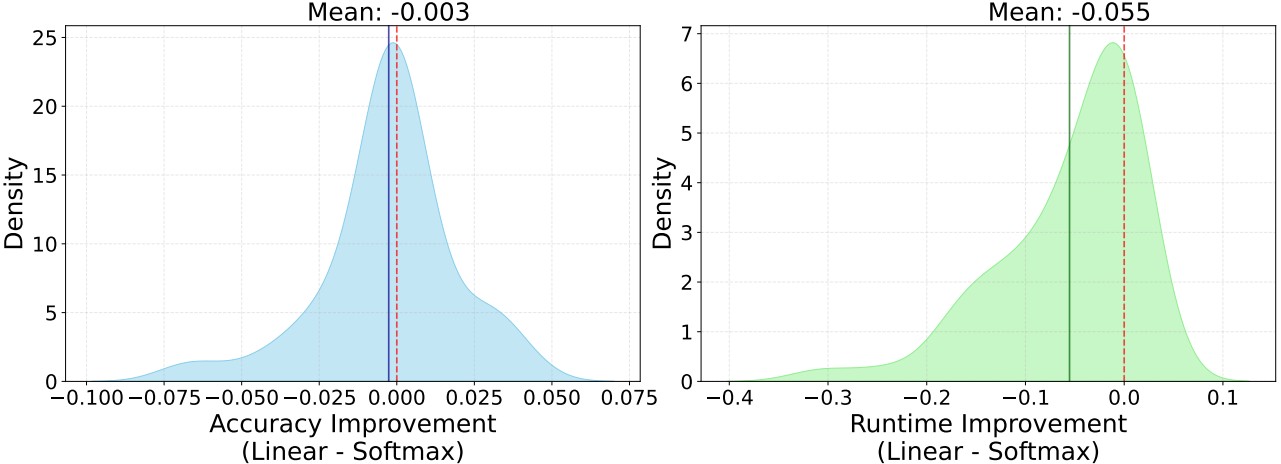

(c) **Accuracy and runtime comparison of softmax and linear attention**. Replacing softmax with linear attention preserves comparable accuracy while significantly reducing runtime.

*Figure 8.* Impact of model architecture on tabular classification performance. Detailed version of Fig. 2.

## B.2. Supplementary Visualizations

Fig. 8 provides additional visualizations to complement the analysis in Fig. 2. It includes both TabPFN-Causal-Masked and TabPFN-Mamba variants in the settings shown in Fig. 2a and Fig. 2b, as the two experiments highlight different aspects of model behavior. To improve the interpretability of performance differences between softmax and linear attention across datasets, we also include a density plot illustrating the changes in runtime and accuracy when the softmax operator is replaced with linear attention.

## C. Supplement to Section 5: TABFLEX

In this section, we provide detailed insights into TABFLEX, including training procedures, validation dataset selection, and analyses such as the sensitivity of model selection thresholds on final performance.

### C.1. Computation Analysis of Various Attention Mechanism

In this part, we provide a computational analysis of various attention mechanisms, comparing standard attention, FlashAttention (specifically FlashAttention-I (Dao et al., 2022)), causal FlashLinearAttention (referred to as FlashLinearAttention in

Yang et al. (2024)), and non-causal linear attention. To clarify, FlashLinearAttention is designed to reduce HBM access specifically for causal linear attention. For notational simplicity, we use the term "linear attention" to refer to non-causal linear attention. For both linear attention and FlashLinearAttention, we assume an element-wise kernel feature mapping. The assumption is reasonable, as popular kernel feature mappings like $\mathrm{elu}(\cdot) + 1$ (used here) and $\mathrm{ReLU}(\cdot)$ are also element-wise. We denote this mapping as $\phi$.

---

**Algorithm 2** Causal FlashLinearAttention Implementation (Yang et al., 2024)

---

**Input:** Matrices $\boldsymbol{Q}, \boldsymbol{K}, \boldsymbol{V} \in \mathbb{R}^{N \times D}$ in HBM, on-chip SRAM of size $M$

1 Set block size $B$  Initialize $\boldsymbol{O} = (0)_{N \times D} \in \mathbb{R}^{N \times D}$ in HBM  Divide $\boldsymbol{Q}$ into $T = \lceil \frac{N}{B} \rceil$ blocks $\boldsymbol{Q}_1, \ldots, \boldsymbol{Q}_T$ of size $B \times D$ each, and divide $\boldsymbol{K}, \boldsymbol{V}$ into $T = \lceil \frac{N}{B} \rceil$ blocks $\boldsymbol{K}_1, \ldots, \boldsymbol{K}_T$ and $\boldsymbol{V}_1 \ldots \boldsymbol{V}_T$ of size $B \times D$ each  Divide $\boldsymbol{O}$ into $T$ blocks $\boldsymbol{O}_1, \ldots, \boldsymbol{O}_T$ of size $B \times D$ each  On on-chip SRAM, construct causal mask, $\boldsymbol{M} \in \mathbb{R}^{B \times B}$  On SRAM, initialize $\boldsymbol{S} = (0)_{D \times D} \in \mathbb{R}^{D \times D}$  **for** $1 \leq j \leq T$ **do**

2  $\quad$ Load $\boldsymbol{K}_j, \boldsymbol{V}_j, \boldsymbol{Q}_j, \boldsymbol{O}_j$ from HBM to on-chip SRAM  On chip, compute $\boldsymbol{K}_j \leftarrow \phi(\boldsymbol{K}_j)$  On chip, compute $\boldsymbol{Q}_j \leftarrow \phi(\boldsymbol{Q}_j)$  Write $\boldsymbol{O}_j \leftarrow \boldsymbol{Q}_j \boldsymbol{S} + ((\boldsymbol{Q}_j \boldsymbol{K}_j^\top) \odot \boldsymbol{M}) \cdot \boldsymbol{V}_j$ to HBM  On-chip, compute $\boldsymbol{S} \leftarrow \boldsymbol{S} + \boldsymbol{K}_j^\top \boldsymbol{V}_j$

3 **end**

**Output:** $\boldsymbol{O}$

---

We evaluate these mechanisms based on their High Bandwidth Memory (HBM) access, memory requirements, and floating-point operations (FLOP) when computing attention outputs given query, key, and value inputs. While Dao et al. (2022) have provided computations for standard attention and FlashAttention, we focus our analysis on causal FlashLinearAttention (detailed in Alg. 2) and HBM-efficient non-causal linear attention (developed by us and detailed in Alg. 3) in Section C.1.1. In practice, we employ a simplified PyTorch implementation of linear attention and demonstrate its efficiency, as it only causes marginal increases in HBM access and memory usage as we demonstrate in Section C.1.2. Furthermore, we present visualizations in Section C.1.2 that illustrate the time and CUDA memory consumption of these attention mechanisms across various sequence lengths and scenarios.

---

**Algorithm 3** HBM-Efficient Implementation of Linear Attention

---

**Input :** Matrices $\boldsymbol{Q}, \boldsymbol{K}, \boldsymbol{V} \in \mathbb{R}^{N \times D}$ in HBM, on-chip SRAM of size $M$

1 Set block size $B$  Initialize $\boldsymbol{O} = (0)_{N \times D} \in \mathbb{R}^{N \times D}$ in HBM  Divide $\boldsymbol{Q}$ into $T = \lceil \frac{N}{B} \rceil$ blocks $\boldsymbol{Q}_1, \ldots, \boldsymbol{Q}_T$ of size $B \times D$ each, and divide $\boldsymbol{K}, \boldsymbol{V}$ into $T = \lceil \frac{N}{B} \rceil$ blocks $\boldsymbol{K}_1, \ldots, \boldsymbol{K}_T$ and $\boldsymbol{V}_1, \ldots, \boldsymbol{V}_T$ of size $B \times D$ each  Divide $\boldsymbol{O}$ into $T$ blocks $\boldsymbol{O}_1, \ldots, \boldsymbol{O}_T$ of size $B \times D$ each  On on-chip SRAM, initialize $\boldsymbol{S} = (0)_{D \times D} \in \mathbb{R}^{D \times D}$  **for** $1 \leq i \leq T$ **do**

2  $\quad$ Load $\boldsymbol{K}_i, \boldsymbol{V}_i$  On chip, compute $\boldsymbol{K}_i \leftarrow \phi(\boldsymbol{K}_i)$  On chip, compute $\boldsymbol{S} \leftarrow \boldsymbol{S} + \boldsymbol{K}_i^\top \boldsymbol{V}_i$

3 **for** $1 \leq j \leq T$ **do**

4  $\quad$ Load $\boldsymbol{Q}_j, \boldsymbol{O}_j$  On chip, compute $\boldsymbol{Q}_j \leftarrow \phi(\boldsymbol{Q}_j)$  Write $\boldsymbol{O}_j \leftarrow \boldsymbol{Q}_j \boldsymbol{S}$ to HBM

**Output:** $\boldsymbol{O}$

---

### C.1.1. HBM-EFFICIENT LINEAR ATTENTION

Here, we analyze the number of HBM accesses, HBM memory, and FLOPS required by FlashLinearAttention (Alg. 2) and linear attention (Alg. 3).

**Lemma 2.** *Let $\boldsymbol{Q}, \boldsymbol{K}, \boldsymbol{V} \in \mathbb{R}^{N \times D}$ represent the query, key, and value matrices for a single attention head, where $N$ is the sequence length and $D$ is the embedding size. Both FlashLinearAttention (Alg. 2) and linear attention (Alg. 3) require $5ND$ HBM accesses to compute the attention output.*

*Proof of Lemma 2.* For causal FlashLinearAttention (Alg. 2):

- Line 8: Loading $\boldsymbol{K}_j, \boldsymbol{V}_j, \boldsymbol{Q}_j, \boldsymbol{O}_j$ necessitates $4BD$ HBM accesses.

- Line 11: Writing $\boldsymbol{O}_j$ requires $BD$ HBM accesses.

These operations are executed $T$ times, where $T = \lceil \frac{N}{B} \rceil$. Thus, the total HBM accesses are:

$$5BD \cdot T = 5BD \cdot \lceil \frac{N}{B} \rceil = 5ND.$$

For non-causal linear attention (Alg. 3):

- Line 7: Loading $\boldsymbol{K}_i, \boldsymbol{V}_i$ requires $2BD$ HBM accesses.

- Line 11: Loading $\boldsymbol{Q}_j, \boldsymbol{O}_j$ requires $2BD$ HBM accesses.

- Line 13: Writing $\boldsymbol{O}_j$ requires $BD$ HBM accesses.

These operations are also repeated $T$ times, where $T = \lceil \frac{N}{B} \rceil$. Consequently, the total HBM accesses are:

$$5BD \cdot T = 5BD \cdot \lceil \frac{N}{B} \rceil = 5ND.$$

Therefore, we conclude that both causal FlashLinearAttention and non-causal linear attention require $5ND$ HBM accesses to compute the attention output. $\square$

**Lemma 3.** *Let $\boldsymbol{Q}, \boldsymbol{K}, \boldsymbol{V} \in \mathbb{R}^{N \times D}$ represent the query, key, and value matrices for a single attention head, where $N$ is the sequence length and $D$ is the embedding size. Both FlashLinearAttention (Alg. 2) and linear attention (Alg. 3) require $4ND$ HBM memory to compute the attention output.*

*Proof of Lemma 3.* For both algorithms:

- Storing $\boldsymbol{Q}, \boldsymbol{K}, \boldsymbol{V}$ requires $3ND$ memory.

- Storing $\boldsymbol{O}$ requires $ND$ memory.

Total HBM memory usage: $4ND$. $\square$

**Lemma 4.** *Let $\boldsymbol{Q}, \boldsymbol{K}, \boldsymbol{V} \in \mathbb{R}^{N \times D}$ represent the query, key, and value matrices for a single attention head, where $N$ is the sequence length and $D$ is the embedding size. Both FlashLinearAttention (Alg. 2) and linear attention (Alg. 3) require $O(ND^2)$ FLOPS to compute the attention output.*

*Proof of Lemma 4.* For causal FlashLinearAttention (Alg. 2):

- Computing $\phi(\boldsymbol{K}_j)$ and $\phi(\boldsymbol{Q}_j)$ requires $2BD$ FLOPs.

- Computing $(\boldsymbol{Q}_j \boldsymbol{K}_j^\top) \odot \mathbf{M}$ requires $B^2(2D-1) + B^2$ FLOPs.

- The result of last step multiplied by $\boldsymbol{V}_j$ requires $B^2(2D-1) + BD(2B-1)$ FLOPs.

- Computing $\boldsymbol{Q}_j \boldsymbol{S}$ requires $B \cdot D(2D-1)$ FLOPs.

- The addition of the last two steps requires $BD$ FLOPs.

- Computing $\boldsymbol{K}_j^\top \boldsymbol{V}_j$ (line 10) requires $(2B-1) \cdot D^2$ FLOPs.

- The addition of $\boldsymbol{S}$ and the last step requires $D^2$ FLOPs.

The total number of FLOPs for one iteration is:

$$2BD + B^2(2D - 1) + B^2 + B^2(2D - 1) + BD(2B - 1) + B \cdot D(2D - 1) + BD + (2B - 1) \cdot D^2 + D^2$$
$$= 4B^2D + 2BD + 4BD^2.$$

These operations are repeated $T = \lceil \frac{N}{B} \rceil$ times. The total number of FLOPs is:

$$(4B^2D + 2BD + 4BD^2) \cdot T = O(ND^2).$$

For non-causal linear attention (Alg. 3):

- Computing $\phi(\boldsymbol{K}_i)$ requires $BD$ FLOPs.

- Computing $\boldsymbol{S} + \boldsymbol{K}_i^\top \boldsymbol{V}_i$ (line 9) requires $2BD^2$ FLOPs.

- Computing $\phi(\boldsymbol{Q}_j)$ (line 12) requires $BD$ FLOPS.

- Computing $\boldsymbol{Q}_j \boldsymbol{S}$ (line 13) requires $(2D - 1)BD$ FLOPs.

These operations are repeated $T = \lceil \frac{N}{B} \rceil$ times. The total number of FLOPs is:

$$(BD + 4BD^2) \cdot T = O(ND^2).$$

Thus, we conclude that both algorithms require $O(ND^2)$ FLOPs to compute the attention output. $\qquad\square$

### C.1.2. SIMPLIFIED PYTORCH IMPLEMENTATION OF LINEAR ATTENTION

In our implementation, we adopt a straightforward PyTorch approach to linear attention rather than an HBM-efficient method. We employ the concise two-line implementation presented in Listing 1. In the following lemma, we demonstrate that this straightforward implementation only incurs a marginal increase in HBM accesses and HBM memory usage.

```
def linear_attn(q, k, v):
    """
    q: (batch, heads, seq_q, dim_qk)
    k: (batch, heads, seq_kv, dim_qk)
    v: (batch, heads, seq_kv, dim_v)
    """
    kv = torch.einsum("bhnd,bhnm->bhdm", k, v)
    o = torch.einsum("bhld,bhdm->bhlm", q, kv)
    return o.contiguous()
```

*Listing 1.* Straightforward PyTorch implementation of linear attention (Katharopoulos et al., 2020).

**Theorem 1.** *Let $\boldsymbol{Q}, \boldsymbol{K}, \boldsymbol{V} \in \mathbb{R}^{N \times D}$ represent the query, key, and value matrices for a single attention head, where $N$ is the sequence length and $D$ is the embedding size. Both causal FlashLinearAttention (Alg. 2) and non-causal linear attention (Listing 1) require $O(ND)$ HBM accesses, $O(ND)$ HBM memory, and $O(ND^2)$ FLOPS to compute the attention output.*

*Proof.* Let us consider the implementation in Listing 1 and compare it to Alg. 3. PyTorch's optimized tensor computation ensures efficiency, with the primary distinction between Listing 1 and Alg. 3 being the storage of `kv` in the former, which is equivalent to $\boldsymbol{S} \in \mathbb{R}^{D \times D}$ in Alg. 3. This results in the following changes:

- HBM Accesses: By Lemma 2, Alg. 3 requires $5ND$ HBM accesses. Due to the additional write and load operations for $\boldsymbol{S} \in \mathbb{R}^{D \times D}$, Listing 1 requires $5ND + 2D^2$ HBM accesses.

- HBM Memory Usage: By Lemma 3, Alg. 3 requires $4ND$ HBM memory usage. Due to the additional storage requirements for $\boldsymbol{S} \in \mathbb{R}^{D \times D}$, Listing 1 requires $4ND + D^2$ HBM memory usage.

The number of FLOPS remains unaffected. The analysis above, in conjunction with Lemmas 2, 3, and 4, yields the desired outcome. ☐

In Table 5, we summarize the #HBM access, HBM memory, and FLOPS required by standard attention (with naive PyTorch implementation), FlashAttention-I, FlashLinearAttention (causal), and linear attention with both implementations.

| | Standard Attention | FlashAttention (Dao et al., 2022) | FlashLinearAttention (Yang et al., 2024) | Linear Attention Alg. 3 | Linear Attention Listing 1 |
|---|---|---|---|---|---|
| # HBM access | $4N^2 + 4ND$ | $\frac{12N^2D^2}{M} + \frac{16N^2D}{M} + 2ND$ | $5ND$ | $5ND$ | $5ND + 2D^2$ |
| Memory | $2N^2 + 4ND$ | $2N + 4ND$ | $4ND$ | $4ND$ | $4ND + D^2$ |
| FLOPS | $O(N^2D)$ | $O(N^2D)$ | $O(ND^2)$ | $O(ND^2)$ | $O(ND^2)$ |

*Table 5.* **Comparison of memory and computational costs across different attention mechanisms.** FlashAttention improves the speed of standard attention by optimizing # HBM access. Flash causal linear attention takes a similar approach, achieving linear # HBM access. However, we show that non-causal linear attention already achieves linear # HBM access, matching the efficiency of flash causal linear attention without requiring any additional optimization on # HBM access.

Subsequently, we visualize the empirical execution time and CUDA memory utilization of FlashAttention-2, FlashLinearAttention, and linear attention in Fig. 9a and Fig. 9b, respectively. We vary the head dimension $\in \{32, 64, 128, 256\}$, the number of heads $\in \{2, 4, 8, 16\}$, and the sequence length $\in \{2^4, 2^5, \ldots, 2^{15}\}$. We focus on the self-attention case, randomly generating input (serving as key, query, and values) with a batch size of 10, and replicate the experiment 5 times. The final values presented are aggregated across these 5 simulations. Notably, we were unable to obtain results for FlashLinearAttention in two configurations: (1) head dimension 256 with 8 heads, and (2) head dimension 256 with 16 heads, due to illegal memory access error incurred by the PyTorch package `fla` (Yang et al., 2024). Our observations from the figures indicate that both runtime and CUDA memory usage of FlashLinearAttention and linear attention exhibit linear growth with respect to sequence length, aligning with the predictions of Theorem 1.

## C.2. Model Training

We implement linear attention with the feature function $\texttt{elu}(\cdot) + 1$, adhering to the default implementation proposed by Katharopoulos et al. (2020). Unless otherwise specified, we adopt the training setup of TABPFN for TABFLEX-S100, TABFLEX-L100, and TABFLEX-H1K. Each model is trained on a single Nvidia A100 80GB PCIe GPU.

| Hyperparameters | Batch Size | Epoch | Learning Rate | #Steps/epoch |
|---|---|---|---|---|
| TABFLEX-S100 | 1210 | 8 | 3e-5 | 8192 |
| TABFLEX-L100 | 110 | 4 | 3e-5 | 8192 |
| TABFLEX-H1K | 1410 | 4 | 3e-5 | 1024 |

*Table 6.* **Hyperparameters used for training TABFLEX models.** The number of steps per epoch indicates the quantity of synthetic datasets generated and used for training within each epoch.

Table 6 summarizes the hyperparameters selected for training TABFLEX-S100, TABFLEX-L100, and TABFLEX-H1K. For all three methods, we utilize the same embedding size of 512, consistent with TABPFN. We extend the feature capacity by modifying the first linear layer, which projects the features into embeddings – specifically, we increase the number of neurons responsible for receiving the features.

The training loss curves are illustrated in Fig. 10. We observe that as the number of features and the length of training dataset sequences increase, the training process becomes more time-consuming. In fact, training a robust TABFLEX-H1K model requires more than three weeks.

## C.3. Validation Datasets

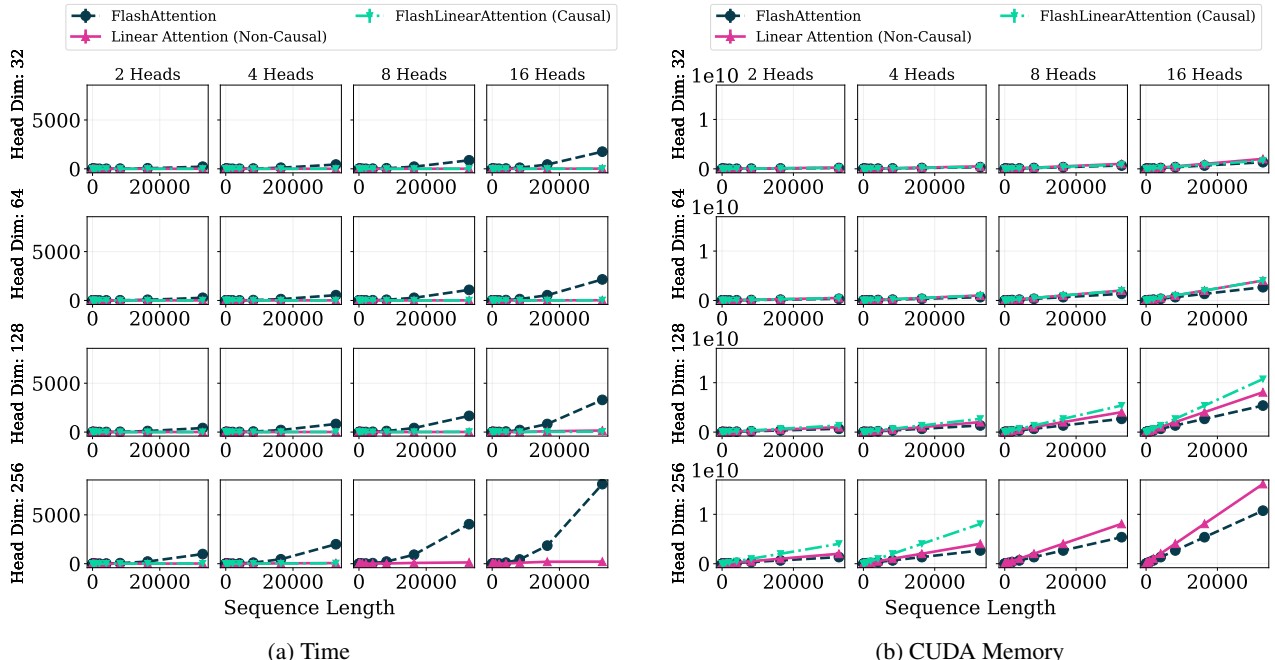

(a) Time                                          (b) CUDA Memory

*Figure 9.* Time and CUDA memory usage comparison of FlashAttention-2 (Dao, 2024), causal FlashLinearAttention (Yang et al., 2024), and linear attention (Katharopoulos et al., 2020) (implemented as in Listing 1). Results for FlashLinearAttention in two configurations: (1) head dimension 256 with 8 heads, and (2) head dimension 256 with 16 heads are missing, due to illegal memory access error incurred by the PyTorch package fla (Yang et al., 2024).

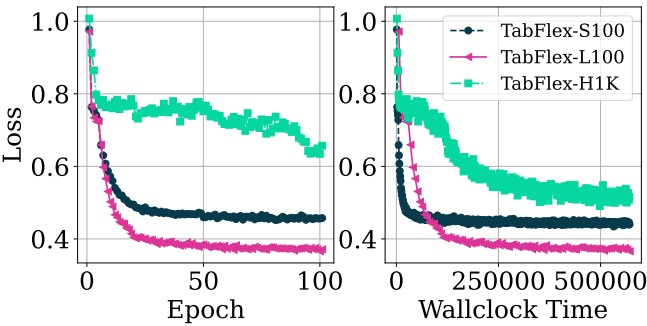

*Figure 10.* Visualization of training loss for TABFLEX models as a function of epoch and wallclock time.

We select the validation datasets from the OpenML AutoML Benchmark (Feurer et al., 2021) by choosing 10 datasets from each of the following sample size intervals: $[0.1K, 1K)$, $[1K, 10K)$, and $[10K, 100K)$. To ensure diversity in the validation set, we also vary the number of classes and features within each interval. The details of all datasets used in validation are summarized in Table 7.

### C.4. Sensitivity Analysis on Decision Boundaries

The decision thresholds align with the training regimes of the models. TABFLEX-S100, sharing TabPFN's training setup but with an updated architecture, is deployed similarly ($n \leq 3K, d \leq 100$). TABFLEX-L100, trained on low-dimensional ($d \leq 100$) but larger datasets, is used for longer sequences ($n \geq 3K, d \leq 100$). TABFLEX-H1K, trained on high-dimensional data, is assigned to handle those cases accordingly.

We note that performance is not highly sensitive to the chosen decision boundaries. To demonstrate this, we conducted additional experiments on simple (low-dimension, small-size), low-dimensional & large, and high-dimensional & large datasets—two datasets per setting—and present the results for all three models in Table 8.

## D. Supplement to Section 6: Performance Evaluation of TABFLEX

### D.1. TabZilla Datasets

The results of our experiments on TabZilla-related datasets are reported in Table 11, 1, and 2. (McElfresh et al., 2023) presents the details of the datasets used in their hard benchmark (Table 2) in Table 4 of their paper. We provide the

| OpenML did | Dataset | #Features | #Instances | #Classes |
|---|---|---|---|---|
| 279 | meta-stream-intervals.arff | 75 | 45164 | 11 |
| 311 | oil-spill | 50 | 937 | 2 |
| 742 | fri-c4-500-100 | 101 | 500 | 2 |
| 825 | boston-corrected | 21 | 506 | 2 |
| 833 | bank32nh | 33 | 8192 | 2 |
| 841 | stock | 10 | 950 | 2 |
| 920 | fri-c2-500-50 | 51 | 500 | 2 |
| 940 | water-treatment | 37 | 527 | 2 |
| 981 | kdd-internet-usage | 69 | 10108 | 2 |
| 1039 | hiva-agnostic | 1618 | 4229 | 2 |
| 1491 | one-hundred-plants-margin | 65 | 1600 | 100 |
| 1492 | one-hundred-plants-shape | 65 | 1600 | 100 |
| 1503 | spoken-arabic-digit | 15 | 263256 | 10 |
| 1515 | micro-mass | 1301 | 571 | 20 |
| 1536 | volcanoes-b6 | 4 | 10130 | 5 |
| 1541 | volcanoes-d4 | 4 | 8654 | 5 |
| 1549 | autoUniv-au6-750 | 41 | 750 | 8 |
| 40645 | GAMETES-Epistasis-2-Way-1000atts-0.4H-EDM-1-EDM-1-1 | 1001 | 1600 | 2 |
| 40672 | fars | 30 | 100968 | 8 |
| 40677 | led24 | 25 | 3200 | 10 |
| 40693 | xd6 | 10 | 973 | 2 |
| 40705 | tokyo1 | 45 | 959 | 2 |
| 40922 | Run-or-walk-information | 7 | 88588 | 2 |
| 40985 | tamilnadu-electricity | 4 | 45781 | 20 |
| 41082 | USPS | 257 | 9298 | 10 |
| 41144 | madeline | 260 | 3140 | 2 |
| 41986 | GTSRB-HOG01 | 1569 | 51839 | 43 |
| 41988 | GTSRB-HOG02 | 1569 | 51839 | 43 |
| 41989 | GTSRB-HOG03 | 2917 | 51839 | 43 |
| 41990 | GTSRB-HueHist | 257 | 51839 | 43 |
| 41991 | Kuzushiji-49 | 785 | 270912 | 49 |
| 42193 | compas-two-years | 14 | 5278 | 2 |
| 42206 | porto-seguro | 38 | 595212 | 2 |
| 42343 | KDD98 | 478 | 82318 | 2 |

*Table 7.* Characteristics of datasets in our diverse validation set.

| Dataset | | Metric | TabPFN | TabFlex-S100 | TabFlex-L100 | TabFlex-H1K |
|---|---|---|---|---|---|---|
| **Simple** | credit-g | Accuracy | 0.79 | **0.82** | 0.79 | 0.75 |
| | | Time (s) | 0.23 | 0.13 | 0.13 | 0.13 |
| | diabet | AUC | **0.78** | **0.78** | 0.77 | **0.78** |
| | | Time (s) | 0.15 | 0.08 | 0.10 | 0.09 |
| **Low-Dimensional & Large** | bank-marketing | AUC | 0.89 | 0.89 | **0.90** | 0.89 |
| | | Time (s) | 1.75 | 0.25 | 2.43 | 1.67 |
| | elevators | AUC | 0.94 | 0.94 | **0.95** | 0.94 |
| | | Time (s) | 1.11 | 0.22 | 0.7 | 0.7 |
| **High-dimensional & Large** | nomao | AUC | 0.86 | 0.83 | 0.75 | **0.99** |
| | | Time (s) | 1.95 | 0.86 | 4.71 | 4.63 |
| | SpeedDating | AUC | 0.66 | 0.69 | 0.59 | **0.83** |
| | | Time (s) | 2.86 | 0.89 | 1.63 | 1.71 |

*Table 8.* Performance of TABFLEX-S100, TABFLEX-L100, and TABFLEX-H1K across three types of datasets, using TabPFN as a baseline. We observe that all TABFLEX variants perform well on both simple and low-dimensional large datasets, demonstrating that performance is fairly robust to the choice of decision threshold.

specifications of the datasets used for our evaluation in Table 11 and Table 1 in Table 9 and Table 10, respectively.

## D.2. Extended Experiment Setup

**Baselines.** We evaluate our approach against a comprehensive set of baselines, as considered by McElfresh et al. (2023). These include: (i) classical methods: Random Forest (Liaw et al., 2002), SVM (Cortes, 1995), LinearModel (Cox, 1958), KNN (Cover & Hart, 1967) and Decision Tree (Quinlan, 1986); (ii) Gradient Boosted Decision Trees (GBDT) methods: XGBoost (Chen & Guestrin, 2016), CatBoost (Prokhorenkova et al., 2018), and LightGBM (Ke et al., 2017); (iii) Non-Transformer Neural Network (Non-TF NN) methods: SAINT (Somepalli et al., 2021), ResNet (He et al., 2016), DANet (Chen et al., 2022), NODE (Popov et al., 2019), MLP (Rumelhart et al., 1986), MLP-rtdl (Gorishniy et al., 2021), DeepFM (Guo et al., 2017), STG (Yamada et al., 2020), VIME (Yoon et al., 2020), and NAM (Agarwal et al., 2021); (iv) Transformer (TF) methods: TABPFN (Hollmann et al., 2023), FTTransformer (Gorishniy et al., 2021), TabNet (Arik & Pfister, 2021), and TabTransformer (Huang et al., 2020). The results for these methods, except TABPFN, are taken directly from McElfresh et al. (2023), who conducted their experiments using a V100 GPU, while our experiments are run on an A100 GPU, which may introduce slight variations in performance. Additionally, we incorporate two recent methods designed for scaling tabular classification: TuneTables (Feuer et al., 2024), a TF method, and HyperFast (Bonet et al., 2024), a Non-TF NN method.

Note that not all baselines successfully ran on all datasets. Many methods face constraints and encounter issues, particularly with the TabZilla hard benchmark, often due to poor scalability. We explicitly indicate which methods failed to run smoothly across all datasets. Originally, TABPFN was limited to datasets with no more than 100 features and 10 classes. To facilitate a fair comparison between TABFLEX and TABPFN, we implemented workarounds to prevent TABPFN from encountering errors. For datasets exceeding 100 features, we performed random feature selection. For those with more than 10 classes, we evaluated the accuracy of the nine most prevalent classes and marked all other classes as other, and incorrect. For TuneTables, we directly import `TuneTablesClassifier` from their Python package `tunetables`. Note that our results differ from those reported in their paper, as their study involved more extensive hyperparameter search, which significantly increased runtime. We also compare our methods with TuneTables using the dataset split specified in their paper's setting, with results deferred to Section D.5. Similarly, for HyperFast, we utilize `HyperFastClassifier` directly from their Python package `hyperfast` default parameters. Notably, HyperFast is meta-trained on many datasets we use for evaluation.

## D.3. Results on 98 Simple Datasets (Table 1, McElfresh et al. (2023))

The results, presented in Table 11, are consistent with the conclusions drawn in the main text.

## D.4. Evaluation on Additional Datasets

We provide additional evaluation of TABFLEX on eight large datasets randomly selected from OpenML-CC18 Benchmarks (Bischl et al., 2019), after excluding the datasets contained in TabZilla's evaluation. As shown in Table 12, TABFLEX consistently outperforms TABPFN in terms of speed and achieves superior performance on the majority of the datasets.

## D.5. Additional Comparison with TuneTables

As mentioned in Section 6, the results of TuneTables presented in Table 13 of our main experiments use `TuneTablesClassifier`. However, we note that the original paper reported results after 30 iterations of hyperparameter tuning. They also applied this process to TABPFN, using a different subset of datasets as training samples at each iteration. In Table 13, we compare the performance of TABFLEX without any hyperparameter tuning to the results reported in their paper. TABFLEX remains competitive, particularly when the number of samples is limited. While TuneTables tends to perform better with larger sample sizes due to its ability to update model parameters based on training data, TABFLEX maintains comparable performance while being significantly faster.

## D.6. Extending TABFLEX for Image Classification

We explore the application of TABFLEX to image classification tasks, comparing it against MLP and ResNet architectures. Our evaluation uses straightforward configurations without extensive hyperparameter optimization to maintain reasonable computational costs. The MLP implementations include both two-layer and three-layer variants, each configured with 10

| Dataset | D | N | C | Dataset | D | N | C | Dataset | D | N | C |
|---|---|---|---|---|---|---|---|---|---|---|---|
| cmc | 9 | 1473 | 3 | socmob | 5 | 1156 | 1 | adult-census | 14 | 32561 | 2 |
| kc1 | 21 | 2109 | 1 | vehicle | 18 | 846 | 4 | breast-cancer | 9 | 286 | 2 |
| kc2 | 21 | 522 | 1 | heart-h | 13 | 294 | 1 | mfeat-factors | 216 | 2000 | 10 |
| pc3 | 37 | 1563 | 1 | jasmine | 144 | 2984 | 1 | mfeat-zernike | 47 | 2000 | 10 |
| pc4 | 37 | 1458 | 1 | phoneme | 5 | 5404 | 1 | dresses-sales | 12 | 500 | 2 |
| pc1 | 21 | 1109 | 1 | semeion | 256 | 1593 | 10 | mfeat-fourier | 76 | 2000 | 10 |
| cjs | 33 | 2796 | 6 | heart-c | 13 | 303 | 1 | balance-scale | 4 | 625 | 3 |
| car | 6 | 1728 | 4 | kr-vs-kp | 36 | 3196 | 1 | bank-marketing | 16 | 45211 | 2 |
| tae | 5 | 151 | 3 | spambase | 57 | 4601 | 1 | car-evaluation | 21 | 1728 | 4 |
| jm1 | 21 | 10885 | 1 | satimage | 36 | 6430 | 6 | cylinder-bands | 37 | 540 | 2 |
| dna | 180 | 3186 | 3 | mushroom | 22 | 8124 | 1 | mfeat-karhunen | 64 | 2000 | 10 |
| musk | 167 | 6598 | 1 | diabetes | 8 | 768 | 1 | credit-approval | 15 | 690 | 2 |
| wdbc | 30 | 569 | 1 | rabe_266 | 2 | 120 | 1 | ozone-level-8hr | 72 | 2534 | 2 |
| wilt | 5 | 4839 | 1 | breast-w | 9 | 699 | 1 | analcatdata_dmft | 4 | 797 | 6 |
| ilpd | 10 | 583 | 1 | elevators | 18 | 16599 | 1 | monks-problems-2 | 6 | 601 | 2 |
| sick | 28 | 3772 | 1 | Satellite | 36 | 5100 | 1 | cardiotocography | 35 | 2126 | 10 |
| iris | 4 | 150 | 3 | fertility | 9 | 100 | 1 | PhishingWebsites | 30 | 11055 | 2 |
| lymph | 18 | 148 | 4 | ionosphere | 34 | 351 | 1 | synthetic_control | 60 | 600 | 6 |
| churn | 20 | 5000 | 1 | transplant | 3 | 131 | 1 | steel-plates-fault | 27 | 1941 | 7 |
| colic | 22 | 368 | 1 | eucalyptus | 19 | 736 | 5 | mfeat-morphological | 6 | 2000 | 10 |
| ecoli | 7 | 336 | 8 | Australian | 14 | 690 | 1 | acute-inflammations | 6 | 120 | 2 |
| autos | 25 | 205 | 6 | hayes-roth | 4 | 160 | 3 | analcatdata_boxing1 | 3 | 120 | 2 |
| scene | 299 | 2407 | 1 | dermatology | 34 | 366 | 6 | analcatdata_chlamydia | 3 | 100 | 2 |
| profb | 9 | 672 | 1 | MiceProtein | 77 | 1080 | 8 | wall-robot-navigation | 24 | 5456 | 4 |
| colic | 26 | 368 | 1 | SpeedDating | 120 | 8378 | 1 | visualizing_livestock | 2 | 130 | 2 |
| labor | 16 | 57 | 1 | tic-tac-toe | 9 | 958 | 1 | Click_prediction_small | 11 | 39948 | 2 |
| irish | 5 | 500 | 1 | hill-valley | 100 | 1212 | 1 | analcatdata_authorship | 70 | 841 | 4 |
| glass | 9 | 214 | 6 | page-blocks | 10 | 5473 | 5 | banknote-authentication | 4 | 1372 | 2 |
| yeast | 8 | 1269 | 4 | lung-cancer | 56 | 32 | 3 | LED-display-domain-7digit | 7 | 500 | 10 |
| sonar | 60 | 208 | 1 | qsar-biodeg | 41 | 1055 | 1 | visualizing-environmental | 3 | 111 | 2 |
| splice | 60 | 3190 | 3 | fri_c3_100_5 | 5 | 100 | 1 | postoperative-patient-data | 8 | 88 | 2 |
| libras | 104 | 360 | 10 | ada_agnostic | 48 | 4562 | 1 | blood-transfusion-service-center | 4 | 748 | 2 |
| anneal | 38 | 898 | 5 | fri_c0_100_5 | 5 | 100 | 1 | | | | |

*Table 9.* Datasets utilized in the evaluation presented in Table 11. Here $D$, $N$, and $C$ denote the number of features, instances, and classes, respectively.

| Dataset | #Features | #Instances | #Classes |
|---|---|---|---|
| Australian | 14 | 690 | 2 |
| LED-display-domain-7digit | 7 | 500 | 10 |
| MiceProtein | 77 | 1080 | 8 |
| acute-inflammations | 6 | 120 | 2 |
| analcatdata_authorship | 70 | 841 | 4 |
| analcatdata_boxing1 | 3 | 120 | 2 |
| analcatdata_chlamydia | 3 | 100 | 2 |
| analcatdata_dmft | 4 | 797 | 6 |
| anneal | 38 | 898 | 5 |
| autos | 25 | 205 | 6 |
| balance-scale | 4 | 625 | 3 |
| blood-transfusion-service-center | 4 | 748 | 2 |
| blood-transfusion-service-center | 4 | 748 | 2 |
| breast-cancer | 9 | 286 | 2 |
| breast-w | 9 | 699 | 2 |
| colic | 26 | 368 | 2 |
| colic | 22 | 368 | 2 |
| credit-approval | 15 | 690 | 2 |
| cylinder-bands | 37 | 540 | 2 |
| dermatology | 34 | 366 | 6 |
| diabetes | 8 | 768 | 2 |
| dresses-sales | 12 | 500 | 2 |
| ecoli | 7 | 336 | 8 |
| eucalyptus | 19 | 736 | 5 |
| fertility | 9 | 100 | 2 |
| fri_c0_100_5 | 5 | 100 | 2 |
| fri_c3_100_5 | 5 | 100 | 2 |
| glass | 9 | 214 | 6 |
| hayes-roth | 4 | 160 | 3 |
| heart-c | 13 | 303 | 2 |
| heart-h | 13 | 294 | 2 |
| hill-valley | 100 | 1212 | 2 |
| ilpd | 10 | 583 | 2 |
| ionosphere | 34 | 351 | 2 |
| iris | 4 | 150 | 3 |
| irish | 5 | 500 | 2 |
| kc2 | 21 | 522 | 2 |
| labor | 16 | 57 | 2 |
| lung-cancer | 56 | 32 | 3 |
| lymph | 18 | 148 | 4 |
| monks-problems-2 | 6 | 601 | 2 |
| pc1 | 21 | 1109 | 2 |
| postoperative-patient-data | 8 | 88 | 2 |
| profb | 9 | 672 | 2 |
| qsar-biodeg | 41 | 1055 | 2 |
| rabe_266 | 2 | 120 | 2 |
| socmob | 5 | 1156 | 2 |
| sonar | 60 | 208 | 2 |
| synthetic_control | 60 | 600 | 6 |
| tae | 5 | 151 | 3 |
| tic-tac-toe | 9 | 958 | 2 |
| transplant | 3 | 131 | 2 |
| vehicle | 18 | 846 | 4 |
| visualizing_environmental | 3 | 111 | 2 |
| visualizing_livestock | 2 | 130 | 2 |
| wdbc | 30 | 569 | 2 |
| yeast | 8 | 1269 | 4 |

*Table 10.* Datasets utilized in the evaluation presented in Table 1.

| Algorithm | Class | Mean AUC | | Std. AUC | | Time / 1000 inst. | |
|---|---|---|---|---|---|---|---|
| | | median | mean | mean | median | median | mean |
| TabPFN (Hollmann et al., 2023) | TF | 0.97 | 0.84 | 0.15 | 0.08 | 0.56 | 0.74 |
| CatBoost (Prokhorenkova et al., 2018) | GBDT | 0.97 | 0.92 | 0.15 | 0.07 | 1.95 | 20.51 |
| TABFLEX (Ours) | TF | 0.96 | 0.90 | 0.15 | 0.08 | 0.22 | 0.37 |
| XGBoost (Chen & Guestrin, 2016) | GBDT | 0.96 | 0.91 | 0.16 | 0.09 | 0.38 | 0.85 |
| RandomForest (Liaw et al., 2002) | Classical | 0.95 | 0.90 | 0.16 | 0.09 | 0.32 | 0.47 |
| SAINT (Somepalli et al., 2021) | TF | 0.94 | 0.86 | 0.16 | 0.11 | 146.15 | 170.56 |
| HyperFast (Bonet et al., 2024) | Non-TF NN | 0.94 | 0.87 | 0.15 | 0.09 | 53.45 | 89.75 |
| LightGBM (Ke et al., 2017) | GBDT | 0.93 | 0.85 | 0.18 | 0.09 | 0.29 | 0.90 |
| ResNet (He et al., 2016) | Non-TF NN | 0.93 | 0.85 | 0.16 | 0.10 | 8.83 | 15.99 |
| DANet (Chen et al., 2022) | Non-TF NN | 0.92 | 0.85 | 0.16 | 0.08 | 57.18 | 64.29 |
| NODE (Popov et al., 2019) | Non-TF NN | 0.91 | 0.83 | 0.16 | 0.11 | 131.73 | 160.76 |
| FTTransformer (Gorishniy et al., 2021) | TF | 0.89 | 0.81 | 0.17 | 0.11 | 18.04 | 27.91 |
| SVM (Cortes, 1995) | Classical | 0.89 | 0.78 | 0.19 | 0.09 | 2.06 | 61.18 |
| MLP-rtdl (Gorishniy et al., 2021) | Non-TF NN | 0.88 | 0.75 | 0.18 | 0.11 | 7.09 | 15.21 |
| DeepFM (Guo et al., 2017) | Non-TF NN | 0.87 | 0.77 | 0.19 | 0.12 | 4.89 | 6.05 |
| TabNet (Arik & Pfister, 2021) | TF | 0.85 | 0.68 | 0.26 | 0.14 | 29.34 | 35.12 |
| STG (Yamada et al., 2020) | Non-TF NN | 0.82 | 0.71 | 0.20 | 0.14 | 15.98 | 18.58 |
| TuneTables (Feuer et al., 2024) | TF | 0.81 | 0.70 | 0.25 | 0.16 | 32.96 | 73.40 |
| LinearModel (Cox, 1958) | Classical | 0.78 | 0.67 | 0.19 | 0.14 | 0.03 | 0.04 |
| MLP (Rumelhart et al., 1986) | Non-TF NN | 0.76 | 0.68 | 0.20 | 0.13 | 11.23 | 18.31 |
| DecisionTree (Quinlan, 1986) | Classical | 0.74 | 0.63 | 0.24 | 0.18 | 0.01 | 0.03 |
| TabTransformer (Huang et al., 2020) | TF | 0.72 | 0.61 | 0.17 | 0.13 | 13.45 | 22.05 |
| KNN (Cover & Hart, 1967) | Classical | 0.70 | 0.61 | 0.21 | 0.14 | 0.03 | 0.05 |
| VIME (Yoon et al., 2020) | Non-TF NN | 0.60 | 0.54 | 0.25 | 0.15 | 15.60 | 17.98 |
| NAM (Agarwal et al., 2021) | Non-TF NN | 0.39 | 0.44 | 0.27 | 0.19 | 97.99 | 233.77 |

*Table 11.* **Performance comparison of algorithms across 98 simple datasets (as used in Table 1 of McElfresh et al. (2023)).** The reported AUC values are normalized. The "Time/1000 inst." column represents the combined training and test time for all datasets, divided by the total number of samples. Notably, TABFLEX achieves top 3 performance, with faster runtimes compared to baselines of similar performance, and a $2\times$ speedup relative to TABPFN.

hidden neurons and trained for 70 epochs at a fixed learning rate of 0.001. The ResNet architecture employs 2 residual blocks with main and hidden dimension sizes of 128 and 256, respectively. The experimental results demonstrate that TABFLEX achieves remarkable efficiency gains, operating $30\times$ faster than the MLP and $400\times$ faster than the ResNet while maintaining competitive performance. This represents a significant advancement in image classification efficiency, particularly noteworthy given that previous approaches like TABPFN were constrained to small, low-dimensional datasets.

We further evaluate TABFLEX on an 10-way classification CIFAR-10 image dataset (Krizhevsky et al., 2009) of 60K samples for which each sample is a color image of $32 \times 32$ size. We deploy two approaches to convert images to 1D vectors. First, we flatten the RGB channels to obtain a vector of 3072 dimensions. Second, we utilize a pretrained ResNet-18 (He et al., 2016) to obtain a semantically meaningful representation of the image as an 384-dim vector. For each approach, we then feed the 1D vectors to TABFLEX as a tabular dataset. As shown in Table 14, with first approach (CIFAR10-flattened), we achieve an AUC of 0.791 within seconds of inference. We find that reducing feature dimension to 30% with PCA leads to 4 times lower latency while preserving the AUC score. The second approach (CIFAR10-embedding) significantly increases AUC to 92.2% while reducing the inference latency by six times.

# E. Supplement to Section 7: Ablation Studies

## E.1. Datasets for Section 7.2: Incorporating Data-Efficient Techniques

For dimensionality reduction, we use the following datasets from OpenML: dna, musk, scene, jasmine, semeion, Speed-Dating, hill-valley, mfeat-factors. These datasets are selected from Table 9 where feature dimensions are greater than 100. For the random sampling experiment, we use the following datasets from OpenML: cmc, kc1, car, yeast, car-evaluation, mfeat-morphological, mfeat-zernike, banknote-authentication, socmob. The tested datasets are selected from Table 9 where the data size is greater than 1000 instances, and the feature dimension is lower than 100.

| Dataset | #Features | #Instances | #Classes | Mean AUC | | Mean Time (seconds) | |
|---|---|---|---|---|---|---|---|
| | | | | TABPFN | TABFLEX | TABPFN | TABFLEX |
| kick | 33 | 72983 | 2 | 0.663 | **0.684** | 13.330 | **3.096** |
| Click-prediction-small-1220 | 10 | 39948 | 2 | 0.652 | **0.659** | 3.663 | **0.887** |
| house-8L | 9 | 22784 | 2 | **0.947** | 0.945 | 1.383 | **0.536** |
| okcupid-stem | 20 | 50789 | 3 | 0.825 | **0.828** | 6.152 | **1.511** |
| volcanoes-b1 | 4 | 10176 | 5 | 0.660 | **0.663** | 0.349 | **0.202** |
| volcanoes-b2 | 4 | 10668 | 5 | 0.651 | **0.652** | 0.375 | **0.217** |
| kdd-internet-usage | 69 | 10108 | 2 | **0.932** | **0.932** | 1.021 | **0.851** |
| BNG(tic-tac-toe) | 10 | 39366 | 2 | **0.836** | 0.835 | 3.626 | **1.111** |

*Table 12.* **Performance comparison between TABPFN and TABFLEX on an additional large dataset.** We observe that TABFLEX is consistently faster than TABPFN and outperforms it on the majority of the datasets.

| Dataset | Size | TABPFN | | TuneTables | | TABFLEX | |
|---|---|---|---|---|---|---|---|
| | | Acc. | Runtime (sec.) | Acc. | Runtime (sec.) | Acc. | Runtime (sec.) |
| breast-cancer | 286 | .765 | 29 | .770 | 65 | **.793** | 1 |
| heart-c | 303 | .848 | 40 | **.903** | 66 | **.903** | 0 |
| ecoli | 336 | .848 | 30 | .843 | 66 | **.882** | 0 |
| colic | 368 | .856 | 39 | **.892** | 66 | **.892** | 0 |
| dresses-sales | 500 | .578 | 41 | **.580** | 122 | **.580** | 0 |
| cylinder-bands | 540 | .800 | 41 | **.846** | 82 | .796 | 0 |
| climate | 540 | .959 | 59 | .951 | 97 | **.963** | 0 |
| balance-scale | 625 | .990 | 29 | .995 | 55 | **1.000** | 0 |
| blood-transfusion | 748 | .801 | 25 | .782 | 56 | **.840** | 0 |
| cmc | 1473 | .554 | 91 | .556 | 109 | **.605** | 0 |
| kc-1 | 2109 | .862 | 168 | .856 | 187 | **.867** | 0 |
| bioresponse | 3151 | .797 | 638 | **.798** | 3012 | .720 | 13 |
| christine | 5418 | .742 | 666 | **.755** | 3920 | .721 | 11 |
| robert | 10000 | .250 | 964 | **.414** | 2397 | .333 | 17 |
| dilbert | 10000 | .922 | 761 | **.992** | 3749 | .802 | 17 |
| har | 10299 | .936 | 370 | **.981** | 2657 | .918 | 9 |
| eeg-eye-state | 14980 | .940 | 178 | **.986** | 1929 | .837 | 1 |
| elevators | 16599 | .902 | 186 | .902 | 1297 | **.907** | 1 |
| riccardo | 20000 | .922 | 1395 | **.995** | 5247 | .773 | 31 |
| volkert | 58310 | .567 | 459 | **.693** | 6331 | .561 | 12 |
| higgs | 67557 | .671 | 931 | **.714** | 4084 | .691 | 1 |
| connect-4 | 98050 | .668 | 931 | **.817** | 5395 | .692 | 1 |
| BNG (vote) | 131072 | .968 | 1976 | **.974** | 2493 | **.974** | 1 |
| albert | 425240 | .642 | 2363 | **.658** | 17518 | .637 | 1 |
| airlines | 539383 | .600 | 2602 | **.653** | 44434 | .597 | 2 |
| BNG (labor) | 1000000 | .937 | 5518 | **.967** | 7717 | .950 | 8 |
| agrawall | 1000000 | .948 | 5158 | **.950** | 45504 | .948 | 3 |
| poker-hand | 1025009 | .531 | 2423 | **1.000** | 10471 | .542 | 15 |
| click-prediction-small | 1997410 | .833 | 10421 | **.837** | 33148 | .833 | 5 |

*Table 13.* Accuracy comparison of TABPFN, TuneTables, and TABFLEX on test datasets from Feuer et al. (2024). Results for TABPFN and TuneTables are directly sourced from Feuer et al. (2024), where hyperparameter tuning was performed 30 times for both methods. For TABPFN, hyperparameters determine the subset of the dataset used in ICL. TABFLEX results are reported without hyperparameter tuning.

| Dataset | Two-Layer MLP | | Three-Layer MLP | | ResNet | | TABFLEX (Ours) | |
|---|---|---|---|---|---|---|---|---|
| | AUC | Time (s) | AUC | Time (s) | AUC | Time (s) | AUC | Time (s) |
| MNIST | 0.924 | 23.547 (30.5×) | 0.959 | 23.060 (29.9×) | - | - | 0.948 | 0.771 |
| Fashion-MNIST | 0.793 | 23.340 (28.8×) | 0.853 | 23.604 (29.1×) | .990 | 398.45 (491.1×) | 0.979 | 0.810 |
| CIFAR-10 (flattened) | - | - | - | - | - | - | 0.791 | 5.872 |
| CIFAR-10 (embedding) | - | - | - | - | - | - | 0.922 | 0.989 |

*Table 14.* Performance comparison of TABFLEX against baseline models on image datasets. *\*Note: MLP and ResNet require significantly more time for training and inference, compared to* TABFLEX. *Missing evaluation will be provided in the supplementary.*

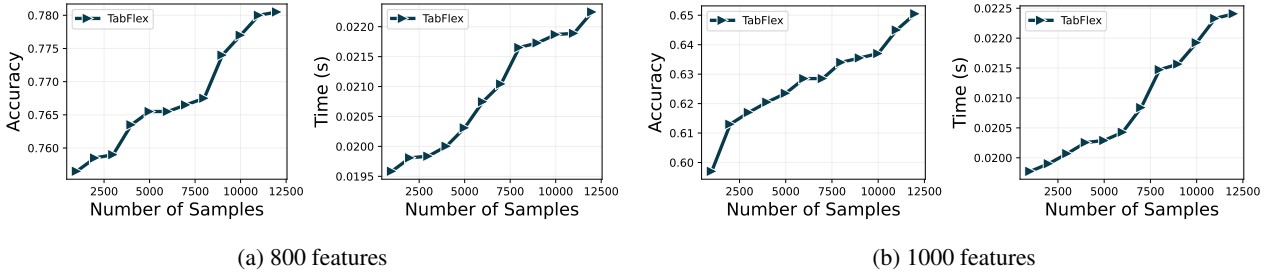

(a) 800 features       (b) 1000 features

*Figure 11.* **Accuracy and runtime versus the number of samples.** Two settings are considered: (a) 800 features and (b) 1000 features. Each curve is averaged over 20 synthetic datasets with varying data distributions, generated the same algorithm as employed in TabPFN (Hollmann et al., 2023).

## E.2. Performance and Runtime vs. Training Sample Size

We have demonstrated that TABFLEX performs well across diverse tasks. Here, we provide a more fine-grained analysis, examining how performance and runtime vary with the number of training samples. Following the setup of TABPFN (Hollmann et al., 2023), we generate synthetic datasets with sample sizes ranging from 1,000 to 12,000 and feature dimensions of 800 and 1,000. Results are averaged over 20 synthetic datasets and presented in Fig. 11. We observe that accuracy consistently improves with more samples, while runtime increases linearly with the sample size, regardless of feature count.

