# OpenReview forum: "TabFlex: Scaling Tabular Learning to Millions with Linear Attention"
_ICML.cc/2025/Conference — ICML 2025 spotlightposter_

### Official Review · Reviewer_1peU · 2025-03-09

**Overall Recommendation:** 3

**Summary:**

This paper explores the use of linear attention for TabPFN, to overcome its limitations in terms of scalability. Indeed, TabPFN is limited by the quadratic complexity of self-attention, making it inefficient for datasets with more than a few 1,000s samples. The authors:
- Demonstrate experimentally that linear attention is preferable to causal linear attention as in State-Space models.
- Introduce TabFlex, a TabPFN variant with linear attention, comprising three models trained on datasets of varying sizes (up to 1,152 vs. 50K samples, 100 vs. 1K features, and 10 vs. 100 classes). TabFlex selects the appropriate model based on dataset size and class count.
- Evaluate TabFlex on a benchmark of small datasets (98 + 57 datasets), showing it matches TabPFN's performance—already the best among many baselines including boosted trees—while achieving a 2× speedup.
- Evaluate TabFlex on the Tabzilla hard benchmark (36 datasets) where it remains on par with TabPFN, offering 2× speedup. It is however inferior to other models including XGBoost, albeit slightly faster.
- Evaluate TabFlex on vectorized CIFAR-10 images.
- Show that feature and sample downsampling can further improve TabFLEX efficiency without affecting performances.

**Claims And Evidence:**

Yes, claims are supported by clear and convincing evidence.

**Essential References Not Discussed:**

The following recent papers address TabPFN’s scalability challenges using approaches different from linear attention. Positioning this work in relation to these references would provide useful context.

[1] Xu et al 2024, Mixture of In-Context Prompters for Tabular PFNs.

[2] Thomas et al 2024, Retrieval & Fine-Tuning for In-Context Tabular Models.

[3] Ma et al 2024, In-Context Data Distillation with TabPFN.

**Experimental Designs Or Analyses:**

The experimental designs are sound.

**Methods And Evaluation Criteria:**

The methods and evaluation criteria are well chosen.

**Other Comments Or Suggestions:**

**Typos**

L.173 With more samples are provided => When

L. 261: reported in respectively reported in Tables 1 and 2

L. 420: relatively well performance

**Minor comments**

* In the impact statement: “Our approach enables an efficient solution for utilizing LLMs” => as TabPFN and TabFlex are not LLMs, I think that the vocabulary should be updated to avoid confusions.

* In Figure 2, it would be nice to have both TabPFN-causal-masked and TABPFN-Mamba on both Figure2a and Figure2b, as both experiments (in Fig. 2a and Fig. 2b) are interesting and do not reveal the same information.

* It is difficult to see from Fig 2c the overall accuracy difference between Softmax attention and Linear attention across all datasets. May be a scatterplot delta_runtime vs delta_accuracy could improve the readability.

* It would be helpful to clarify the specifics of the benchmarks chosen in this paper, such as explaining roughly what simple and hard benchmarks mean.

* On fig. 6 about data subsampling: I am a bit surprised that there is to little improvement after using 20% of the training data. Maybe looking at fraction of training data rather than absolute number of samples hides the signal, i.e, may be having 3,000 samples helps compared to 1,000 samples, but 10,000 instead of 5,000 not so much.

**Other Strengths And Weaknesses:**

**Strengths**
* I truly enjoyed reading this paper: it is very well written, very pedagogical, and its claims are well supported by the experiments.

* Given the limitations of TabPFN, exploring whether linear attention could enhance scalability without significant performance loss is a natural direction. This paper provides a rigorous answer.

* The finding that linear attention tends to preserve performance compared to the original self-attention in TabPFN is interesting.

**Weaknesses**
* A limitation of TabFlex is that in harder cases (may be for larger datasets in particular), it does not provide state-of-the-art performances. In Fig. 4, TabFlex seems significantly outperformed by XGBoost, and overall does not compare favourably to the 5th best method in Table 2. It indicates that scaling the training to larger datasets thanks to linear attention is not enough to obtain sota performance.

* The paper does not analyze performance and speedup as a function of the number of training samples. Since TabFlex replaces the quadratic complexity of TabPFN with linear complexity via linear attention, one would expect the speedup to scale with $n$, the number of samples. It would be helpful to highlight this experimentally.

Moreover, linear attention enables TabFlex to train on larger datasets, suggesting that its performance could improve relative to TabPFN as the dataset size increases. However, the current hard benchmark combines large datasets with high-dimensional and other challenging datasets, making it difficult to isolate this effect. A more targeted analysis of how TabPFN, TabFlex, and other baselines such as XGBoost perform as a function of $n$ would be valuable.

* Since TabFlex consists of three models, a natural question is whether it could be pre-trained across all dataset sizes simultaneously, allowing a single model to perform well in all cases instead of using three separate models. Have the authors explored this possibility?

* I am not convinced of the significance of the results on image classification experiments. The performances should be evaluated in relation to the state-of-the-art (i.e. vision architectures) as fast inference time alone may not be useful if it comes at the cost of significantly lower accuracy.

* Since TabPFN has now been significantly outperformed by TabPFN2 [4] (released in January 2025), a natural question is how incorporating linear attention would impact TabPFN2.

[4] Hollmann et al, Accurate predictions on small data with a tabular foundation model

**Questions For Authors:**

See weaknesses.

In particular the second point on analyzing the effects of the number of samples on performance and speed-up.

**Relation To Broader Scientific Literature:**

This paper investigates whether TabPFN’s scalability limitations can be addressed with linear attention while preserving performance.

**Theoretical Claims:**

There is one theoretical claim comparing the computational and memory complexity of causal FlashLinearAttention and non-causal linear attention.  I did not check the correctness of this claim.

---

> ### Author Rebuttal · Authors · 2025-04-01
>
> We thank Reviewer 1peU for the insightful feedback and constructive suggestions. We are very excited that you truly enjoyed reading our work and finds that (i) our claims are well supported by clear and convincing evidence on well-chosen evaluation criteria, providing interesting finding on performance perservation, (ii) our method is 'natural' and sound with rigorous analysis, (iii) our paper is very well-written. Please find our answers to your comments and questions as follows.
>
>
> > **Comment**: TabFlex does not provide SoTA performance in harder cases (e.g., larger datasets), suggesting that linear attention scaling alone may be insufficient, especially compared to XGBoost.
>
> Thanks for raising this point.
>
> While our goal is not to surpass SoTA but to improve the efficiency-performance trade-off over TabPFN, we ran additional experiments comparing TabFlex with XGBoost on 20 synthetic datasets (from PFN priors).
>
>
> **`Experiment results`**: https://shorturl.at/tH02z
>
> TabFlex outperforms XGBoost in both accuracy and inference speed when the feature count is below 1K. However, for >1K features, while TabFlex remains faster, it lags in accuracy, especially on the real-world datasets used in the paper. We believe this is due to:
> * (i) Training limitation: TabFlex is pretrained on datasets with up to 1K features and struggles to generalize beyond that range.
> * (ii) Distribution shift: PFN-generated synthetic datasets may not reflect real-world distributions, leading to slightly degraded performance at real datasets.
>
> Nevertheless, TabFlex offers a great balance between computational cost and performance in many practical scenarios. We now explicitly include this discussion in the revised paper, emphasizing TabFlex's practical efficiency-performance trade-off.
>
> > **Suggestion**: Analyze performance and speedup as a function of the number of training samples.
>
> Great suggestion.
> We evaluate TabFlex on a set of synthetic 1000-feature datasets generated from prior distributions developed by PFN.
> We vary the number of training samples and visualize the change in accuracy and inference latency.
>
> **`Experiment results`**:  https://shorturl.at/dWap2
>
> As the number of training samples increases, the accuracy indeed improves, and the inference latency (or speed) linearly scales to the number of training samples, as expected.
>
>
>
> > **Q**: Will pre-training across all dataset sizes simultaneously allow a single model to perform well in all cases instead of using three separate models?
>
> Great question. For TabFlex-H1K, we do pre-train across all dataset sizes simultaneously to leverage the full sample set. However, this significantly slows down training (see Fig. 8 in the Appendix), and even after convergence, the model slightly underperforms compared to TabFlex-S100 on small datasets (see  https://shorturl.at/HFaOD). This trade-off is why we opted for an ensemble approach with thresholds based on each model’s specialized training regime.
>
> > **Q**: How would incorporating linear attention impact TabPFNv2?
>
> TabPFNv2 improves TabPFN's performance in the low-sample regime (<10K examples). Since our work targets scalability—an orthogonal aspect—extending our approach to TabPFNv2 could potentially yield benefits in both scalability and performance. We agree this is a promising and interesting direction, and we will add this point to the discussion as suggested.
>
>
> > **Comment**: The performances on image classification should be evaluated with the SoTA (i.e., vision architectures) to highlight accuracy-latency trade-off.
>
> We clarify that image classification is not the focus of our work. The evaluation was intended to explore TabFlex’s adaptability beyond tabular data. We agree that comparing with SoTA vision models would better highlight the limitations of tabular approaches and will note this as a future direction.
>
>
> > **Suggested references**.
>
> Thanks for the suggestion.
> MixturePFN (Xu et al 2024) improves scalability by routing new test samples to a pool of scalable prompters using Sparse Mixture of In-Context Prompters, while LoCalPFN (Thomas et al 2024) proposes retrieving a local subset of task-specific data for efficiently fine-tuning on.
> Ma et al 2024 introduce in-context data distillation to optimize TabPFN’s context and remove the data size constraint.
> We integrated this discussion into our revision.
>
> > **Minor comments**:
> - Typos: We revised the typos
> - Impact statement: We revised to clarify the confusion.
> - Fig.2: thanks, we updated Fig. 2.
> - benchmarks: we added the explanation.
> - Fig. 6: We vary the fraction as a common choice for the datasets of different sizes. We added visualization over the sample size to provide a clearer signal.
>
> ---
> **Final Note:** Thanks again for your insight and thoughtful suggestions. We hope that our responses and new experiment results help you better appreciate our work as well as support accepting our paper.

---

### Official Review · Reviewer_K5HQ · 2025-03-12

**Overall Recommendation:** 4

**Summary:**

The paper proposes an in-context learning architecture for tabular learning. The overall method is incremental to TabPFN.

**Claims And Evidence:**

- **TabFlex improves scalability over TABPFN by using linear attention instead of quadratic attention.**: This claim is well supported by their theoretical analysis aligns with empirical results.
- **TabFlex generalizes image classification tasks well.**: I don't really think this claim is well supported since it only contains MLP and ResNet and only two classes of very simple image datasets. I suggest the authors remove this section from the main text or move it to the appendix.
- **Non-causal attention generally outperforms causal attention**: the author just empirically analyzes this idea. Is there any theory or existing work to support that claim?
- **Mamba vs. Transformer** and **Softmax Attention vs. Linear Attention**: there is also lack of mathematically analysis but that should be fine. If more analysis put on the main text it will be better.

**Essential References Not Discussed:**

Some most recent tabular deep learning methods are not mentioned and discussed, such as [1], [2], [3]

[1] Gorishniy, Yury, et al. "Tabr: Tabular deep learning meets nearest neighbors in 2023." arXiv preprint arXiv:2307.14338 (2023).
[2] Xu, Chenwei, et al. "Bishop: Bi-directional cellular learning for tabular data with generalized sparse modern hopfield model." arXiv preprint arXiv:2404.03830 (2024).
[3] Zhu, Bingzhao, et al. "Xtab: Cross-table pretraining for tabular transformers." arXiv preprint arXiv:2305.06090 (2023).

**Experimental Designs Or Analyses:**

The experimental design and analysis do not seems to have any issues since it follows the most popular tabular benchmarks.

**Methods And Evaluation Criteria:**

- The proposed methods make sense in the tabular learning area. The method use linear attention to reduce the cost, conditional model selection for different dataset sizes and feature dimensions (but the decision thresholds seem somewhat arbitrary).
- The evaluation is very good and includes a lot of datasets and benchmarks. The main issue is lacking of regression tasks even the claim of this paper is on classification.

**Other Comments Or Suggestions:**

Just remove or move the image classification task from main text or to the appendix.

**Other Strengths And Weaknesses:**

I don't see other strengths and weaknesses.

**Questions For Authors:**

I don't have questions.

**Relation To Broader Scientific Literature:**

The paper's method is an incremental to the previous method: TabPFN and shows the potential goodness of to the tabular learning field.

**Theoretical Claims:**

The theoretical claim of **Linear Attention Reduces Complexity** seems good to me.
However, the kernel feature mapping lacks of good explanation.

---

> ### Author Rebuttal · Authors · 2025-04-01
>
> We thank Reviewer K5HQ for the detailed review and constructive suggestions.
> We greatly appreciate that Reviewer K5HQ found that (i) our main claim (on improving scalability over TABPFN by linear attention) is well-supported by both theoretical analysis and corresponding empirical evaluation, (ii) our method is sensible with (iii) very good evaluation on large benchmarks and good theoretical claim. We are also encouraged that the Reviewer finds our supplementary material good and well-designed.
> Please find our answers to your comments and questions as follows.
>
> ---
>
> > **Comment**: The theoretical claim seems good, but the kernel feature mapping lacks a good explanation.
>
> Thank you for the comment. Theorem 1 originally does not assume any kernel feature mapping. We update it to incorporate elementwise kernel mappings (commonly used, e.g., $\text{elu}(\cdot) + 1$), while the original statement still holds. Since the mapping is applied elementwise after loading queries and keys, it adds no extra HBM access or memory, and the additional $2DN$ FLOPS is negligible compared to the total $O(ND^2)$ FLOPS.
>
> See **`Extended version of Theorem 1`** here: https://shorturl.at/MFFwF
>
> > **Comment**: The decision thresholds (of conditional model selection) seem somewhat arbitrary.
>
> The decision thresholds for model selection are **not arbitrary but aligned with the training regimes of the models**. TabFlex-S100, sharing TabPFN’s training setup but with an updated architecture, is deployed similarly $(n ≤ 3K, d ≤ 100)$. TabFlex-L100, trained on low-dimensional $(d ≤ 100)$ but larger datasets, is used for longer sequences $(n \geq 3K, d ≤ 100)$. TabFlex-H1K, trained on high-dimensional data, is assigned to handle those cases accordingly.
>
> We note that **performance is not highly sensitive to the chosen decision boundaries**. To demonstrate this, we conducted *additional experiments* on simple (low-dimension, small-size), low-dimensional & large, and high-dimensional & large datasets—two datasets per setting.
> Our new results for all three models, presented in this table **https://shorturl.at/HFaOD** demonstrate our claim.
>
> > **Comment**: The claim *TabFlex generalizes image classification tasks well* is not well supported since it only contains MLP and ResNet and only two classes of very simple image datasets. Remove this section or move it to the appendix.
>
> Based on your comment, we have moved the results to the Appendix and clarified that image classification is not a core claim. Our goal was to explore TabFlex’s adaptability beyond tabular data; the 10-class MNIST and CIFAR10 results offer preliminary support for this, now framed as a side observation.
>
> > **Suggested References**.
>
> Based on your suggestion, we added the following discussion.
> TabR proposes a retrieval-augmented model with a custom kNN-like component to retrieve and extract signals from the nearest neighbors.
> BiSHop establishes interconnected directional learning modules to process data column-wise and row-wise for
> tabular learning.
> XTab utilizes independent featurizers and federated learning to resolve inconsistent column types and quantities.
>
>
> > **Q**: Is there any theory or existing work to support that non-causal attention generally outperforms causal attention?
>
> A few empirical works [1,2] support our observation that causal attention is suboptimal in the ICL setting. For theoretical work, there is no direct comparison, but most of the theoretical work on ICL is based on non-causal attention [3,4]. We will add them into related works.
>
>
> > **Comment**: The evaluation lacks regression tasks, though the claim of this paper is on classification.
>
> Thanks for the comment. We update our limitation discussion to acknowledge this: "For tabular tasks, our current focus is limited to classification. A simple workaround for regression is to discretize the target range into bins and treat it as a classification problem. An interesting future work is extending TabFlex to regression tasks with a more principled approach involving training on regression-specific synthetic data."
>
>
> > **Comment**: Mamba vs. Transformer and Softmax Attention vs. Linear Attention: More analysis into the main text?
>
> We do not include additional theoretical analysis due to the analytical complexity. However, we recognize this as an interesting future direction and will add it to the discussion section.
>
> ---
> *References*:
> * [1] Ding, Nan, et al. "CausalLM is not optimal for in-context learning." ICLR (2024).
> * [2] Gong, Zhuocheng, et al. "Improving Input-label Mapping with Demonstration Replay for In-context Learning." arXiv (2023).
> * [3] Ahn, Kwangjun, et al. "Transformers learn to implement preconditioned gradient descent for in-context learning." NeurIPS (2023).
> * [4] Bai, Yu, et al. "Transformers as statisticians: Provable in-context learning with in-context algorithm selection." NeurIPS  (2023).
>
> **Final notes**: Thanks again for your positive feedback and thoughtful suggestions.

---

> > ### Comment · Reviewer_K5HQ · 2025-04-01
> >
> > Thanks for your response. Please consider adding the regression tasks or even running a few experiments to prove it really works in the regression settings either in the rebuttal period or once your paper gets accepted following the same setting of"why do tree-based models still outperform deep learning on tabular data".
> >
> > My overall recommendation is reasonable good for the paper, and I will keep my score as it is.
> >
> > Good luck.

---

> > > ### Author Response · Authors · 2025-04-07
> > >
> > > Thank you for carefully reading our response and for your support toward the paper’s acceptance. We also sincerely appreciate your suggestion regarding regression tasks—it’s a valuable direction that can further strengthen our work.
> > >
> > > Below, we include results on regression datasets with numerical features from [1], as suggested by the reviewer. We discretized the targets into 10 and 100 bins uniformly and selected the setting that performed better. As baselines, we used linear regression and XGBoost Regressor (100 estimators, max depth 6), both with default parameters from the Sklearn package. While regression is not the primary focus of TabFlex, we observe that it performs reasonably well. We will include these results in the final version, in response to the reviewer’s insightful feedback.
> > >
> > > | Dataset                 | TabFlex | Linear Regression | XGBoost |
> > > |-------------------------|---------|-------------------|---------|
> > > | cpu_act                 | 0.9622  | 0.7661            | 0.9872  |
> > > | pol                     | 0.7770  | 0.4471            | 0.9876  |
> > > | elevators               | 0.7386  | 0.8336            | 0.8984  |
> > > | wine_quality            | 0.1966  | 0.2842            | 0.4398  |
> > > | Ailerons                | 0.7284  | 0.8137            | 0.8272  |
> > > | houses                  | 0.6803  | 0.6496            | 0.8469  |
> > > | house_16H               | 0.2519  | 0.1708            | 0.5276  |
> > > | diamonds                | 0.9085  | 0.9213            | 0.9477  |
> > > | Brazilian_houses        | 0.8943  | 0.3459            | 0.9828  |
> > > | Bike_Sharing_Demand     | 0.3796  | 0.3291            | 0.6995  |
> > > | nyc-taxi-green-dec-2016 | 0.1547  | 0.3109            | 0.5732  |
> > > | house_sales             | 0.6656  | 0.7375            | 0.8732  |
> > > | sulfur                  | 0.4026  | 0.3068            | 0.7497  |
> > > | medical_charges         | 0.8173  | 0.8118            | 0.9790  |
> > > | MiamiHousing2016        | 0.8112  | 0.7302            | 0.9306  |
> > > | superconduct            | 0.6867  | 0.7169            | 0.9086  |
> > > | yprop_4_1               | 0.0000  | 0.0449            | 0.0000  |
> > > | abalone                 | 0.3689  | 0.4622            | 0.5125  |
> > >
> > > [1] Grinsztajn, Léo, Edouard Oyallon, and Gaël Varoquaux. "Why do tree-based models still outperform deep learning on typical tabular data?." Advances in neural information processing systems 35 (2022): 507-520.

---

### Official Review · Reviewer_qdGi · 2025-03-20

**Overall Recommendation:** 4

**Summary:**

This paper evaluates scaling the  tabular in-context model TabPFN to larger dataset sizes by using linear attention to circumvent the quadratic memory complexity of regular attention. They first compare to state-space-models (SSMs) like MAMBA, finding SSMs to underperform, attributed to their causal nature (instead of the permutation-equivariant nature of regular or linear attention). They find linear-attention TabPFN to retain most if the predictive accuracy of TabPFN on a large benchmark, while substantially (> 2 times) speeding up the prediction time and allowing to apply TabPFN to substantially larger datasets up to a million datapoints.

**Claims And Evidence:**

In general the claims regarding linear attention and its performance for TabPFN seeem well-supported, although the random subsampling of 3000 for TabPFN seems a bit arbitrary, a subanalysis looking at a range of subset sizes may be interesting here.
[Update: seems 3000 is common practice]

"Notably, TABFLEX is faster and achieves better performance than TABPFN, and is faster than XGBoost while sacrificing only a small margin of performance." -> To me, this does not look like a small margin, this seems like a very substantial margin, and worth discussing potential reasons for it as well.
[Update: authors provided some analysis]

**Essential References Not Discussed:**

TabPFNv2  (https://www.nature.com/articles/s41586-024-08328-6) should be mentioned as concurrent work somewhere.
[Update: promised to be done]

**Experimental Designs Or Analyses:**

As written,  evaluate on well-established benchmarks which make sense.

**Methods And Evaluation Criteria:**

Yes the benchmark datasets make sense, as written above 3000 seems a bit arbitrary. [Update: seems 3000 is common practice]

**Other Comments Or Suggestions:**

None

**Other Strengths And Weaknesses:**

None

**Questions For Authors:**

None

**Relation To Broader Scientific Literature:**

Insights into how well in-context tabular methods can be scaled, in terms of providing results for one particular way to do so.

**Theoretical Claims:**

No theoretical claims.

---

> ### Author Rebuttal · Authors · 2025-04-01
>
> We thank Reviewer qdGi for the constructive feedback and suggestions. We are encouraged that Reviewer qdGi found that our main claim (on linear attention and its performance) is well-supported by a sensible experiment design on the well-established benchmark. Please find our answers to your comments as follows.
>
> ---
>
> > **Comment**: The performance margin between TabFlex and XGBoost seems substantial and worth discussing potential reasons.
>
> Thank you for the suggestion. First, to better understand the regimes where TabFlex and XGBoost outperform each other, we conducted synthetic experiments with feature counts of [100, 200, 400, 600, 800, 1000], generating 20 datasets per count using PFN priors [3].
>
> **`Experiment results`**:  Accuracy-runtime tradeoff curves are visualized in the following figure: https://anonymous.4open.science/r/icml25_rebuttal_tabflex-7455/Figure1_tablfex_vs_xgboost.png
>
> TabFlex outperforms XGBoost in both accuracy and inference speed when the feature count is below 1K. However, for >1K features, while TabFlex remains faster, it lags in accuracy, especially on the real-world datasets used in the paper. We believe this is due to:
>
> * (i) Training limitation: TabFlex is pretrained on datasets with up to 1K features and struggles to generalize beyond that range.
> * (ii) Distribution shift: PFN-generated synthetic datasets may not reflect real-world distributions, leading to slightly degraded performance at real datasets.
>
> Nevertheless, we note that TabFlex offers a great balance between computational cost and performance in many practical scenarios. We now explicitly include this discussion in the revised paper, emphasizing TabFlex's practical efficiency-performance trade-off.
>
>
> > **Suggested references**: TabPFNv2 should be mentioned as concurrent work somewhere.
>
> Thank you for the suggestion. TabPFNv2 [4] improves TabPFN’s performance in the low-data regime (fewer than 10,000 samples), which is complementary to our focus on speed and scalability. We’ve added a mention of TabPFNv2 in the updated version as a promising direction for future work that could be combined with our approach.
>
> > **Comment**: No theoretical claim.
>
> Thanks for the comment. We do include Theorem 1 (page 5) that supports the high bandwidth memory efficiency of linear attention deployed in our proposed method, demonstrating that the straightforward implementation of linear attention achieves linear HBM access, matching the performance of FlashLinearAttention after optimization.
>
> > **Comment**: Subsampling of 3000 for TabPFN seems a bit arbitrary.
>
> Thanks for the comment. We follow the TabZilla framework [1], which recommends subsampling to 3000 due to the quadratic scaling of TabPFN's runtime and memory usage. This setting is commonly adopted [2] for benchmarking TabPFN across diverse datasets.
>
> ---
>
> *References:*
>
> [1] McElfresh, Duncan, et al. "When do neural nets outperform boosted trees on tabular data?" NeurIPS (2023).
>
> [2] Feuer, Benjamin, et al. "Tunetables: Context optimization for scalable prior-data fitted networks." NeurIPS (2024).
>
> [3] Müller, Samuel, et al. "Transformers can do bayesian inference." ICLR (2022).
>
> [4] Hollmann, Noah, et al. "Accurate predictions on small data with a tabular foundation model." Nature (2025).
>
> **Final Note**: We hope that these answers will allay any concerns about our work and convince the reviewer that it will be a welcome contribution to the ICML community. If there are additional questions that we can address to further support our case, please let us know.

---

> > ### Comment · Reviewer_qdGi · 2025-04-01
> >
> > Thanks for your elaboration of the 3000 choice and the promised inclusion of a note for TabPFNv2. Thanks for the clarification with regard to the theorem, missed that. "no theoretical claim" was also *not* meant as a criticism in any way in any case.
> > Before I update my review, I still would caution with regard to the xgboost comparison, does setting depth and number of estimators to 1 really make any sense? It is fine if your model underperforms xgboost in some scenarios results just needs to be clearly described...

---

> > > ### Author Response · Authors · 2025-04-02
> > >
> > > Thank you for your thoughtful feedback.
> > >
> > > We initially set both the number of estimators and depth to 1 to explore whether we could make the tradeoff curves between XGBoost and TabFlex more comparable. However, even with such settings, the curves remained far apart due to TabFlex’s significantly faster inference.
> > >
> > > That said, we appreciate your great point and have updated the XGBoost configuration to 20 estimators and a max depth of 3.
> > > Here is our result: https://anonymous.4open.science/r/icml25_rebuttal_tabflex-7455/XGBoost_depth3_20estimators.png
> > > * The overall trend remains similar—TabFlex outperforms XGBoost in both accuracy and runtime up to 600 features. As dimensionality increases, XGBoost catches up and surpasses TabFlex at 800 features. Still, TabFlex maintains a better overall tradeoff.
> > >
> > > Regarding the case when XGBoost outperforms TabFlex, as described in the rebuttal, we acknowledged and studied these cases, e.g., the number of features is high (> 1000) and vice versa (when TabFlex is better). We have integrated the discussion and will describe in detail these cases in our updated version. Thank you for your suggestion.

---

### Decision · Program_Chairs · 2025-05-01

**Decision:**

Accept (spotlight poster)

**Comment:**

This submission investigates making the TabPFN-class of models more scalable. Itshows that a linear attention is preferrable to state space models and scales well the framework.  The submission led to interest and discussion with the reviewers, which appreciated the gain in scalability and the empricial study. The work should be situated with regards to the TabPFNv2 paper which came out a bit before the deadline and markedly improves upon the original TabPFN.